# Allosteric regulation of lysosomal enzyme recognition by the cation-independent mannose 6-phosphate receptor

Linda J. Olson [1✉], Sandeep K. Misra [2], Mayumi Ishihara[3], Kevin P. Battaile [4,6], Oliver C. Grant [3], Amika Sood[3], Robert J. Woods[3], Jung-Ja P. Kim [1], Michael Tiemeyer[3], Gang Ren [5], Joshua S. Sharp[2] & Nancy M. Dahms[1✉]

The cation-independent mannose 6-phosphate receptor (CI-MPR, IGF2 receptor or CD222), is a multifunctional glycoprotein required for normal development. Through the receptor's ability to bind unrelated extracellular and intracellular ligands, it participates in numerous functions including protein trafficking, lysosomal biogenesis, and regulation of cell growth. Clinically, endogenous CI-MPR delivers infused recombinant enzymes to lysosomes in the treatment of lysosomal storage diseases. Although four of the 15 domains comprising CI-MPR's extracellular region bind phosphorylated glycans on lysosomal enzymes, knowledge of how CI-MPR interacts with ~60 different lysosomal enzymes is limited. Here, we show by electron microscopy and hydroxyl radical protein footprinting that the N-terminal region of CI-MPR undergoes dynamic conformational changes as a consequence of ligand binding and different pH conditions. These data, coupled with X-ray crystallography, surface plasmon resonance and molecular modeling, allow us to propose a model explaining how high-affinity carbohydrate binding is achieved through allosteric domain cooperativity.

[1] Department of Biochemistry, Medical College of Wisconsin, Milwaukee, WI 53226, USA. [2] Department of BioMolecular Sciences, University of Mississippi, Oxford, MS 38677, USA. [3] Complex Carbohydrate Research Center, University of Georgia, Athens, GA 30602, USA. [4] IMCA-CAT, Hauptman-Woodward Medical Research Institute, Argonne, IL, USA. [5] The Molecular Foundry, Lawrence Berkeley National Laboratory, Berkeley, CA 94720, USA. [6]Present address: New York Structural Biology Center, New York City, NY 10027, USA. ✉email: lolson@mcw.edu; ndahms@mcw.edu

 1

Lysosomes are acidified organelles that carry out degradative metabolism critical to many endocytic, phagocytic, and autophagic processes[1,2]. This diverse degradative capacity depends on a collection of over 60 different soluble proteases, glycosidases, nucleases, and lipases. Delivery of these newly synthesized hydrolytic enzymes to lysosomes depends on the P-type lectins, 300-kDa cation-independent mannose 6-phosphate receptor (CI-MPR) and 46-kDa cation-dependent MPR (CD-MPR), that bind the carbohydrate determinant, mannose 6-phosphate (M6P), on lysosomal enzymes. CI-MPR, which is the primary receptor responsible for this intracellular trafficking[1], binds multiple ligands[3] at the cell surface that include M6P-containing cytokines[4] and non-M6P-containing molecules (e.g., insulin-like growth factor 2 (IGF2)[5], plasminogen[6], urokinase-type plasminogen activator receptor (uPAR)) to mediate CI-MPR's roles as a tumor suppressor[7] and regulator of cell growth and differentiation[8]. Given these many functions, it is not surprising that mice lacking the CI-MPR gene die at birth[9,10].

Mutations in lysosomal proteins (mainly enzymes) that result in defective catabolism and substrate accumulation cause lysosomal storage diseases (LSDs). A characteristic of this family of ~70 LSDs is their progressive and debilitating nature due to their impact on multiple organ systems. Treatment is symptomatic for most LSDs, with only 11 having FDA-approved therapies. For example, deficiency of palmitoyl-protein thioesterase 1 (PPT1) causes the fatal neurodegenerative disorder infantile neuronal ceroid lipofuscinosis, and there are no FDA-approved treatments for these infants[11]. CI-MPR's ability to internalize recombinant M6P-containing enzymes delivered to patients by bi-weekly intravenous infusion forms the basis of enzyme replacement therapy for nine of these FDA-approved therapies[12,13]. Despite CI-MPR's critical function in supplying lysosomes with hydrolases and its role in human therapies, knowledge of how CI-MPR interacts with a heterogeneous population of ~60 different lysosomal enzymes is lacking. Furthermore, no structure of CI-MPR or CD-MPR in complex with an enzyme is available.

Many lectins bind sugars by simultaneously engaging multiple sugar-binding sites, termed carbohydrate recognition domains (CRDs). These CRDs are located on (1) a single polypeptide chain (tandem repeats of CRDs) or (2) different polypeptide chains (hetero-oligomers or clustering of monomers on the cell surface). The resulting multivalent interactions significantly increase ligand affinity[14]. CI-MPR's extracytoplasmic region contains 15 homologous domains called "mannose 6-phosphate receptor homology" domains (MRH) due to their similar size (~150 residues) and conserved residues, including disulfide bonding[15]. CI-MPR has four non-adjacent CRDs (domains 3, 5, 9, and 15), each with distinctive phosphomonoester and phosphodiester glycan preferences[16–18]. Our crystal and NMR structures of MRH domains 1–3 and 5 of CI-MPR[19,20], the structures solved by Jones et al. of domains 11–14[21], along with the recently published cryo-EM structure of endogenous CI-MPR[22] reveal that each MRH domain has a similar β-barrel fold. Furthermore, although domain–domain interactions have been shown to stabilize the binding site for IGF2 (domain 11)[23] and M6P (domain 3)[19], how these interactions influence the function and overall structural dynamics of CI-MPR is not fully understood.

We now report the crystal structures of the N-terminal five domains of human CI-MPR, revealing the orientation of two CRDs (domains 3 and 5) with respect to each other at pH 5.5 and 7.0. Analyses of the receptor bound to PPT1 by small-angle X-ray scattering (SAXS) and hydroxyl radical protein footprinting (HRPF) and under different pH conditions (e.g., Golgi, late endosome) support binding and pH-induced conformational change. In addition, negative-staining electron microscopy (EM) images indicate that CI-MPR adopts multiple conformations influenced by M6P binding. Furthermore, quantitative binding measurements, coupled with biophysical analyses, support allosteric regulation of the two CRDs.

## Results

**Effects of ligand binding on domain orientation**. Of the 15 MRH domains, we focused on the N-terminal five domains of CI-MPR containing two CRDs and the interaction site of plasminogen and uPAR (Fig. 1a). Crystallization screening of human CI-MPR domains 1–5 protein in the presence of 10-mM M6P resulted in two conditions yielding diffraction quality crystals. Comparison of the two structures, one obtained at ~pH 5.5 (2.5 Å, PDB 6P8I) and the other at pH ~7.0 (2.8 Å, PDB 6V02), reveals the same domain orientations relative to one another, an inverted "T" (Fig. 1b and Table 1). Interestingly, both conditions show evidence of the *N*-glycan at N591 of a crystallographic neighbor partially occupying the carbohydrate-binding site of domain 5. The pH 5.5 structure had sufficient occupancy to allow carbohydrate refinement while the corresponding region in the pH 7.0 structure did not (Fig. 1b). The other CRD (domain 3) remains unoccupied in both structures, unlike bovine CI-MPR domains 1–3, where we showed domain 3 was bound to a ligand: either M6P or the oligosaccharide of a crystallographic neighbor (Fig. 1c, inset)[24,25]. Although we were surprised that the domain-3 binding pocket did not contain M6P at pH 7.0 conditions, the presence of the *N*-linked glycan near the binding site of domain 5 could perturb conformational equilibria between substructures that stabilize a domain arrangement most favorable for crystallographic packing. Due to the higher resolution and greater degree of completeness, we focus our analyses on the pH ~5.5 structure (PDB 6P8I). The current structure of domains 1–5 allows us to evaluate the consequence of carbohydrate binding to a single CRD on individual domain structures and the overall positioning of domains relative to one another.

A comparison of domains 1–3 of PDB 1SYO (ligand-bound domain 3) to the N-terminal three domains of the current structure of human domains 1–5 (PDB 6P8I, ligand-free domain 3, ligand-bound domain 5) reveals individual domains retain their overall core structure (r.m.s.d. < 0.5 Å). However, a substantial change in the quaternary structure occurs, with an ~45° rotation and 34-Å movement (S386 in loop C) of domain 3 (Fig. 1c). The interdomain linker between domains 2 and 3 adopts a more extended structure in the absence of ligand, allowing for the repositioning of domain 3 (Fig. 1c). The ability to alter conformations of this 9-residue linker region appears to be an essential factor in the relocation of domain 3, and its important role is supported by the high species conservation of amino acids (ConSurf[26]) within this region (Supplementary Fig. 1). In the absence of a ligand bound to domain 3, its C-terminal face contacts the C-terminal face of domain 1 (Fig. 1e). However, upon ligand binding to domain 3, this domain straddles the C-terminal face of domain 1 and the N-terminal face of domain 2 (Fig. 1e). The position of domain 2 relative to domain 1 does not experience a comparable change as domains 1 and 2 overlay closely (Fig. 1c). Together, these findings show that the presence of ligand alters the nature of the relationship of domain 3 to its neighboring domains illustrating the dynamic nature of this N-terminal region of CI-MPR.

**Comparison of structures of the two P-type lectins**. CI-MPR and CD-MPR are the sole members of the P-type lectin family[2]. CD-MPR is a homodimer with a single MRH domain per polypeptide. CD-MPR transitions in a scissoring-like motion from a closed (binding sites closer together), smaller dimer interface to an open (larger distance between binding sites) conformation in the presence of M6P (Supplementary Fig. 2a)[27]. These

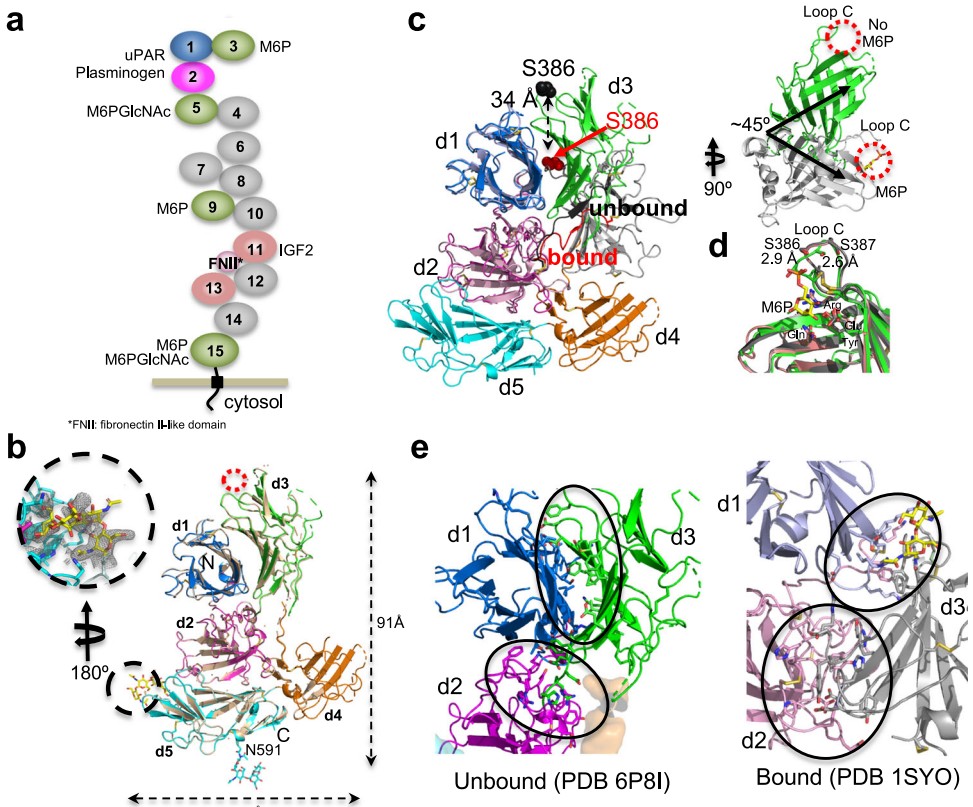

**Fig. 1 The structure of the N-terminal domains of CI-MPR in the presence and absence of ligand. a** Cartoon of the domain structure of CI-MPR, highlighting the multifunctionality of the protein. Relevant ligands are listed next to their known domain of interaction. Domains with no known function to date are colored gray. **b** Overlay of Cα atoms (r.m.s.d. of ~0.2 Å over 524 Cα atoms) of models of crystal structures solved at pH 5.5 (PDB 6P8I) and 7.0 (tan) (PDB 6V02). MRH domains are labeled (d1-d5) along with the N- and C-termini. The red circle marks the unoccupied known M6P binding site in domain 3, while the glycan of a crystallographic neighbor occupying the known binding site of domain 5 (PDB 6P8I) is circled in black. The covalently attached glycan at N591 is shown in stick representation. Approximate dimensions of the domains 1–5 model are given. Inset shows the 2fo-fc map (contoured to ~0.7$\sigma$) around N591 of a crystallographic neighbor. **c** Overlay of the Cα atoms of domain 1 of PDB 1SYO (M6P bound domain 3) (gray) with those of PDB 6P8I (unbound) (green). Residue S386 of loop C is labeled in the unbound (black spheres) and bound (red spheres) structures. The linker between domains 2 and 3 of PDB 6P8I is shown in black while that of 1SYO is shown in red. The rotated inset shows only domain 3 of each structure. The carbohydrate-binding sites are circled in red. The bound M6P (PDB 1SYO) is shown in yellow sticks. **d** A comparison of the known domain 3 structures bound to oligosaccharide of neighbor (pink, PDB 1Q25), M6P (protein by gray cartoon and M6P in yellow sticks, PDB 1SYO), or unbound (green, PDB 6P8I). Displacement of the atoms S386 (2.9 Å) and S387 (2.6 Å) of loop C in the absence of ligand is shown. The four residues essential for carbohydrate binding are labeled. **e** The change in regions of side-chain interactions between domain 3 and domains 1 and 2 upon ligand binding to domain 3 is circled in black.

movements triggered by ligand binding increase the size of the interface and add salt bridges, but create an energetically less favorable association (Complex Formation Significance Score as calculated by PISA[28]) (Supplementary Fig. 2b). In contrast, domain 3 of the multidomain CI-MPR has a composite interface made up of interactions with domains 1 and 2. Rearrangement of domain 3 upon binding ligand reorients its C-terminal β-sheet facilitating communications with both domains 1 and 2, altering salt bridge contacts and reducing the interface (Fig. 1e and Supplementary Fig. 2b).

Despite these two receptors differences, multidomain versus homodimeric, they have significant similarities in their manner of ligand binding. Both receptors use residues located on flexible loops to generate salt bridges with the neighboring domain(s) and hydrophobic cores comprised of residues within the C-terminal β-sheet. CD-MPR maintains a back-to-back (C-terminal β-sheet of one monomer against the C-terminal β-sheet of the other monomer) arrangement in both ligand-bound and unbound states (Supplementary Fig. 2a, c), whereas the domain 1 to domain 3 back-to-back arrangement of CI-MPR is found only in the ligand-unbound state (Fig. 1c).

**Conformation of CI-MPR bound to a lysosomal enzyme.** Because we were unable to obtain crystals of CI-MPR domains 1–5 with an alternate ligand binding scenario, we turned to SAXS. This technique allowed us to gather information on overall protein structure and the effects of (1) ligand binding to domain 3, and (2) ligand absence in domain 5. Although SAXS provides lower resolution data and represents a global average of structures in solution, it has proven to be a robust method to explore biomolecular shapes as well as conformational changes under physiological conditions[29]. We collected SAXS data at pH 6.5 in the absence and presence of M6P or PPT1, both of which interact preferentially with the phosphomonoester-specific binding site of domain 3[17,30]. Data from the domains 1–5 protein produced hyperbolic Kratky plots, characteristic of globular proteins with some tailing indicative of the presence of protein flexibility (Supplementary Fig. 3a). The curves did not significantly change in the presence of either ligand (Supplementary Fig. 3a). In addition, SAXS data, whether collected in the presence or absence of ligand, produce a Porod exponent ($P_E$) of 2.8–2.9 that is indicative of a flexible protein with perhaps some intrinsic disorder (Supplementary Fig. 3b, c)[29].

**Table 1 Data collection and refinement statistics (molecular replacement).**

| | pH 5.5 (PDB 6P8I) | pH 7.0 (PDB 6V02) |
|---|---|---|
| **Data collection** | | |
| Space group | P 1 21 1 | P 1 21 1 |
| Cell dimensions | | |
| $a$, $b$, $c$ (Å) | 50.8 66.2 123.1 | 50.8 66.2 124.5 |
| $\alpha$, $\beta$, $\gamma$ (°) | 90 99.2 90 | 90 100.5 90 |
| Resolution (Å) | 40.0–2.5 (2.63–2.54) | 35.9–2.80 (2.90–2.80) |
| $R_{merge}$ | 0.072 (0.62) | 0.058 (0.54) |
| $I/\sigma I$ | 18.4 (2.13) | 14.8 (1.87) |
| Completeness (%) | 98.4 (91.0) | 90.5 (42.4) |
| Redundancy | 3.7 (3.4) | 2.0 (2.0) |
| **Refinement** | | |
| Resolution (Å) | 40.0–2.5 (2.63−2.54) | 35.9–2.8 (2.90−2.80) |
| No. reflections | 99,803 (8816) | 36,270 (1693) |
| $R_{work}/R_{free}$ | 0.232 (0.321)/0.298 (0.359) | 0.240 (0.389)/0.308 (0.404) |
| No. atoms | | |
| Protein | 5306 | 4570 |
| Ligand/ion/ carbohydrate | 61 | 14 |
| Water | 28 | 74 |
| $B$-factors | 37.7 | 45.2 |
| Protein | 36.1 | 44.9 |
| Ligand/ion/ carbohydrate | 160.4 | 92.6 |
| Water | 61.2 | 52.6 |
| R.m.s. deviations | | |
| Bond lengths (Å) | 0.007 | 0.007 |
| Bond angles (°) | 1.30 | 1.28 |

We then calculated three-dimensional (3D) ab initio models of domains 1–5 in the absence and presence of ligand and compared them to our crystallographic models. The overall shapes of the calculated envelopes in the presence or absence of M6P are similar (Supplementary Fig. 3d and Fig. 2a), resembling a sock with a dimple near the heel. Because M6P is too small to be identified by SAXS, we also utilized the human lysosomal enzyme PPT1. This recombinant, monomeric 279-residue protein harbors three N-linked glycans, and the crystal structure of a monomeric form has already been determined[31]. PPT1 alone gives rise to an oblong 3D model and its position is easily discernable in complex with domains 1–5 (Fig. 2b, c). Based on its preference for the phosphomonoester binding site of domain 3, the envelope is consistent with PPT1 binding to CI-MPR's domain 3 through its M6P-containing glycans. The molecular weight calculated from the envelope volume (MW, (in kDa) = $V_p$ (in nm³)/1.6[32]) (Supplementary Fig. 3c) supports the 1:1 stoichiometry of the complex. Inspection of the 3D envelopes of domains 1–5 bound either to PPT1 (Fig. 2c) or M6P (Supplementary Fig. 3d) shows lack of molecular model (composite model of human domain 1–5 with ligand bound to domains 3 and 5) in the toe region. This absence of model is consistent with domain 5, and perhaps domain 4, residing in a different location than found in our crystallographic structure (PDB 6P8I). Supporting this notion is the observation that domain 4 in our crystallographic structures appears flexible, as demonstrated by discontinuous density, especially in the pH 7.0 structure (PDB 6V02). The flexibility of domain 4 was also recently demonstrated in the cryo-EM structures published by Wang et al.[22] at the time of this paper's submission. Comparing their bovine domains 4–14 structure at pH 7.4 in the presence of IGF2 with our human domains 1–5 structure at pH 5.5 or 7.0 shows domain 4 rotated ~180° (Supplementary Fig. 3e).

We next used the program MultiFoXS to model possible orientations of domains 5 and 4 to improve the fit of our model to the SAXS scattering curves (Fig. 2d, see "Methods" for details of model generation)[33]. Starting with the simplest scenario, only allowing flexibility of the linker between domains 4 and 5, an improved model was calculated with domain 5 swinging into the unpopulated "toe" region (~60 Å) (Supplementary Fig. 3f). Next, we allowed flexibility between both domains 3 and 4 as well as domains 4 and 5 (Fig. 2e). In this model, domain 5 has translated into the toe of the SAXS envelope accompanied by the relocation of domain 4. Allowing either domain 5 or domains 4 and 5 to be flexible and assume alternate conformations from our crystal structure improved the $\chi^2$ of the model fit to the scattering curve from 6.62 to 1.47/1.97 (Fig. 2d, e). Together, these SAXS data (Fig. 2d, e, Supplementary Fig. 3f) are consistent with our hypothesis that domains 4 and 5 are flexible and exist in alternate conformations in the absence of ligand bound to domain 5.

To examine the extent of motions these domains undergo, we negatively stained domains 1–5 in the presence and absence of M6P at pH 7.4 and imaged the samples by electron microscopy (EM). The reference-free 2D classifications yielded 50 classes of domain arrangements in the absence of M6P (Fig. 3). The diversity in the negatively stained EM images demonstrates how dynamic this region of CI-MPR is in the absence of ligand. Although the binding of M6P to domain 3 reduces the number of classes, indicating an overall reduction in domain mobility upon binding, there are still numerous classes representing multiple conformations of domains 1–5. Together, these SAXS and EM data indicate a high degree of flexibility of CI-MPR's N-terminal five domains. However, when domain 5 tethers to the C-terminal domains 6–15 in the native structure, its mobility may be constrained in the context of the full-length receptor.

**Adjacent domains required to achieve high-affinity binding.** Our previous studies examining individual CRDs demonstrated that they bind specifically to ligand but with a lower affinity than in a multidomain construct. For example, domain 3 bound the lysosomal enzyme β-glucuronidase with ~1000-fold lower affinity than domains 1–3 ($K_D = 500$ versus 0.5 nM)[34,35], which could be explained from our crystal structure of domains 1–3 (PDB 1SYO) where interdomain interactions, particularly with domain 1, stabilize the binding pocket of domain 3[19,24]. In contrast, domains 4 and 5 have a minimal effect on the phosphomonoester-specific binding activity of domain 3 (Supplementary Fig. 4a–c). Results of surface plasmon resonance (SPR) analyses show domains 1–3, domains 1–5, and domains 1–5R688A (mutation of R688 previously shown to eliminate carbohydrate binding to domain 5[30]) have similar affinities toward the lysosomal enzyme PPT1 (Supplementary Fig. 4d–f). PPT1 contains phosphomonoester N-glycans predominantly (Supplementary Fig. 4g–i). Thus, the domains 1–3 construct is sufficient to convey high-affinity binding, and domains 4 and 5 are not required for proper carbohydrate-binding function.

We used SPR in conjunction with the lysosomal enzyme acid α-glucosidase (GAA) (modified to contain only phosphodiesters on its N-glycans) to evaluate domain 5's ability to bind phosphodiesters specifically[30]. The presence of the additional four N-terminal domains significantly increases the affinity of domain 5 for GAA phosphodiester ($K_D = $ ~60 nM) (Fig. 4), which is an ~150-fold higher affinity than we previously showed for a construct encoding domain 5 alone[17]. We reported a similar finding of increased binding affinity (~60-fold) comparing a construct encoding domains 5–9 with that of domain 5 alone[17]. Our NMR solution studies on domain 5 alone[20] demonstrates that this domain is stable and capable of specific ligand

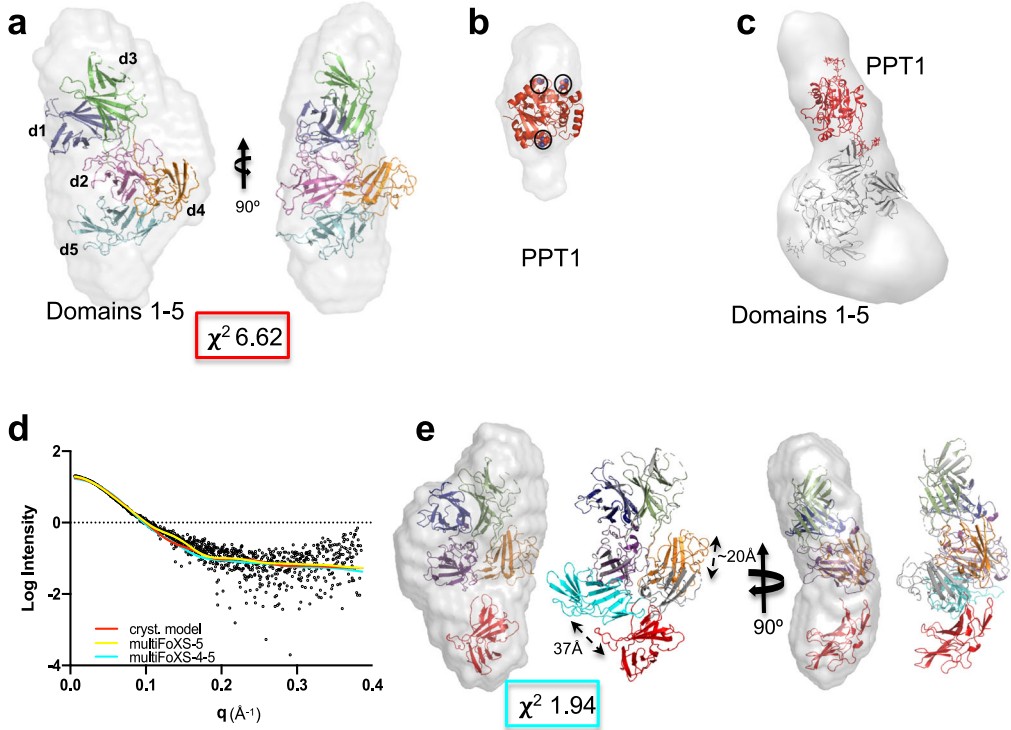

**Fig. 2 Ab initio envelope models rendered as volumes and superimposed onto X-ray crystallographic models. a** X-ray model (PDB 6P8I) placed within envelope derived from SEC-SAXS data of domains 1–5 collected in the absence of **b** PPT1 (PDB 1EI9). The three Asn residues that are glycosylated are shown as spheres (circled in black). **c** The modified X-ray model of domains 1–5 (gray), representing domain 5 in the bound position with the PPT1 model (red) (PDB 1EI9) placed within the calculated ab initio envelope (rendered as a volume illustrating extra density along the most elongated axis). **d** Experimental scattering curve for domains 1–5 (black) overlaid with calculated scattering curves generated from X-ray model (PDB 6P8I, red) or MultiFoXS-generated model based on PDB 6P8I where either the linker between domains 4 and 5 (yellow) or linkers between both domains 3 and 4 and domains 4 and 5 (cyan) are allowed to be flexible. **e** MultiFoXS-4–5 (cyan line in **d**) derived model of domains 1–5 in the absence of ligand placed in the same envelope as in **a** showing relative movements (20 Å) of domains 4 (gray, PDB 6P8I, to orange) and 5 with N682 moving 37 Å (cyan, PDB 6P8I, to red). Corresponding $\chi^2$ value for the curve is shown in **a** and **e**.

recognition. As expected, domains 1–5 bound with high affinity to the lysosomal enzyme GAA modified to contain phosphodiesters or phosphomonoester, and PPT1 (Fig. 4). These data are consistent with the hypothesis that additional domain(s) contribute to the high-affinity binding of domain 5 through protein–protein interactions that serve to stabilize the binding site or through the presence of a secondary binding site(s).

We next evaluated if there were allosteric interactions between CRDs: does carbohydrate binding to one CRD (domain 3) affect ligand binding to the second CRD (domain 5). For these SPR studies, increasing concentrations of domains 1–5 were preincubated with a fixed concentration of PPT1. These solutions were flowed over a sensor chip immobilized with GAA phosphodiester, GAA phosphomonoester, or PPT1. The resulting sensorgrams were analyzed and displayed by double reciprocal plots (Fig. 4d–f). The resulting plots indicate that PPT1 competitively inhibits phosphomonoester ligand binding to domains 1–5. When the receptor-PPT1 complex (PPT1 prebound to domain 3 leaving domain 5 unbound) flowed over a GAA-phosphodiester surface, similar results to those obtained for the phosphomonoester surfaces were observed (Fig. 4) signifying PPT1 can inhibit phosphodiester binding. One explanation is that PPT1 binds to domain 3 and sterically blocks domain 5 from binding GAA phosphodiester. However, this possibility seems unlikely based on SAXS data: we observed the envelope for PPT1 (bound to domain 3) is elongated and points away from the N-terminus and the rest of the receptor (Fig. 2c): PPT1 is not in contact with domain 5. Another possibility is that PPT1 binding to domain 3 causes a rearrangement of

domains such that the binding site of domain 5 is no longer accessible by ligand. This latter possibility of allosteric interactions between the CRDs is consistent with our crystal structures that illustrate that domain interactions can be dramatically altered in the presence or absence of ligands, such as between domain 3 and domains 1 and 2 (Fig. 1c, d). Protein footprinting studies were used to investigate this hypothesis (see below).

**Mapping PPT1 interactions.** To further interrogate receptor-PPT1 interactions, we turned to HRPF utilizing fast photochemical oxidation of proteins (FPOP)[36] as a method to compare protein topography between two structural states (e.g., ligand-bound versus ligand-free). Briefly, proteins are allowed to react with a high concentration of very short-lived hydroxyl radicals generated in situ. Hydroxyl radicals diffuse to the surface of the protein, where they oxidize amino acid side chains forming stable protein oxidation products at the site of oxidation. The rate of this oxidation reaction is directly proportional to the solvent accessible surface area of the amino acid. Changes in amino acid accessibility at the protein surface can be localized and measured by monitoring the rate of reaction of these surface amino acids: occlusion of that portion of the protein surface results in a decrease in the apparent rate of oxidation, while exposure of that portion of the protein surface results in an increase in the rate of oxidation[37]. These stable oxidation products are measured through liquid chromatography-tandem mass spectrometry (LC-MS/MS)[38]. FPOP analysis of the changes in the topography of

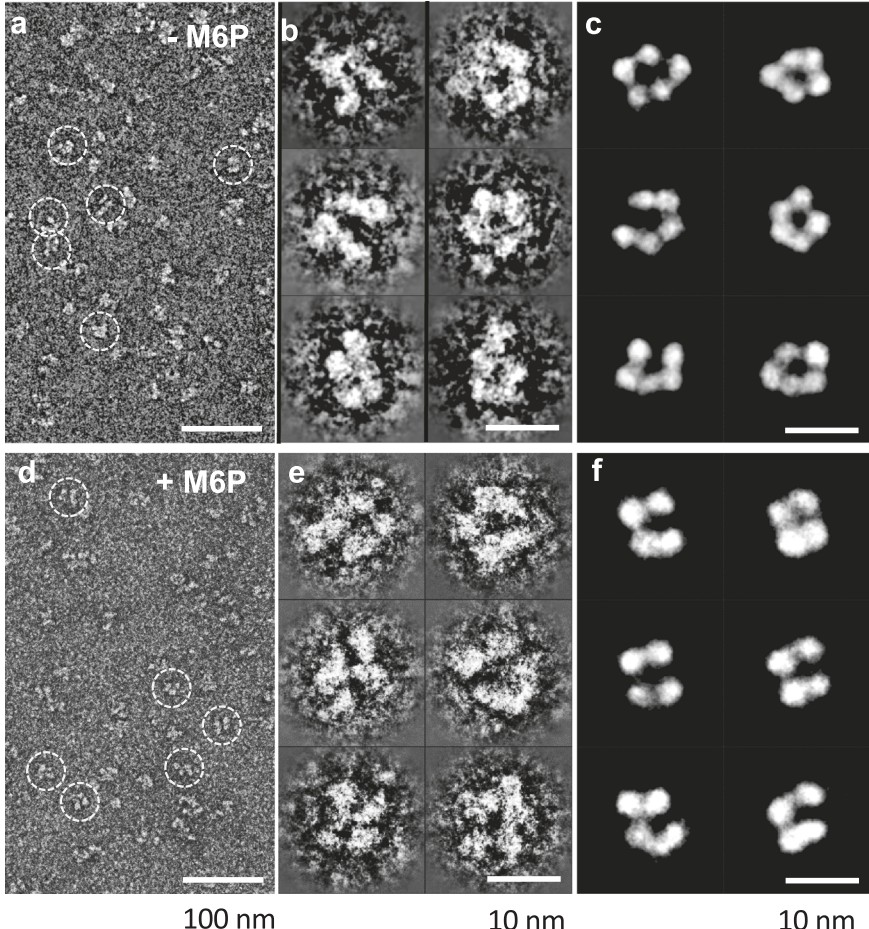

**Fig. 3 Negative-stain electron microscopy of domains 1–5 of CI-MPR in the absence and presence of M6P. a** A survey of a negative-stain TEM image of the sample of CI-MPR domains 1–5 in the absence of 10-mM M6P. **b** Six representative images of reference-free class averages of the particles of CI-MPR domains 1–5 in the absence of 10-mM M6P. **c** Six representative class average images of the particles of CI-MPR domain 1–5 selected from a pool of 35 representative reference-free class averages in the presence of 10-mM M6P. **d** A survey of a negative-stain TEM image of the sample of CI-MPR domains 1–5 in the presence of 10-mM M6P. **e** Six representative images of reference-free class averages of the particles of CI-MPR domains 1–5 in the presence of 10-mM M6P. **f** Six representative class average images of the particles of CI-MPR domain 1–5 selected from a pool of 35 representative reference-free class averages in the presence of 10-mM M6P.

CI-MPR domains 1–5 upon the addition of PPT1 reveals three regions of change distributed over 4 of the 5 domains (Fig. 5a and Supplementary Fig. 5a). PPT1 binding: (1) occludes highly species conserved regions (Supplementary Fig. 1) of the interface between domains 1 and 3, including loop C (previously identified as being part of the high-affinity carbohydrate-binding site) (Fig. 5b[24]), (2) occludes the C-terminal region of front β-sheet (strands 2–4) of domain 5 (Fig. 5c); (3) causes domain 4 to exhibit a topographical rearrangement resulting in occlusion of some surfaces and exposure of others (Fig. 5d and Supplementary Fig. 5b–d). Together, these findings demonstrate a binding-induced conformational change of CI-MPR. Because the interaction of PPT1 with domains 1–5 occurs with a 1:1 stoichiometry (Supplementary Fig. 3c), supported by SPR analyses showing only one CRD can be engaged in ligand binding at a time (Fig. 4), these data are consistent with PPT1's phosphomonoester-containing *N*-glycans binding to domain 3 causing a reorientation (allosteric) of domains such that domain 5 is no longer able to bind ligand (Supplementary Fig. 5d).

**Evidence for a secondary carbohydrate-binding site**. Two peptides in domain 3 experience decreased oxidation rates upon the addition of PPT1, indicating a decrease in their solvent accessibility (Supplementary Fig. 5e–f). Peptide 370–391 is located in the M6P binding site of domain 3 and contains residue R391 that is highly conserved and is essential for high-affinity M6P binding. The observed decreased oxidation rate of peptide 370–391 is consistent with PPT1 binding to this region of the receptor and altering solvent accessibility to hydroxyl radicals. The second peptide, 91–101, is located on β-strand 7 on the C-terminal β sheet of domain 1. As shown in our crystal structures of domains 1–3 bound to carbohydrate, domain 1's β-strand 7 resides across from domain 3's M6P binding site (Supplementary Fig. 5e–f). Although this peptide in domain 1 is outside the known M6P binding region in domain 3, its proximity coupled with its altered oxidation rate upon PPT1 binding raises the possibility it functions as part of a secondary site of carbohydrate interaction. The existence of a secondary, lower affinity binding site is consistent with SPR analyses that showed high and moderate binding affinities for CI-MPR domains 1–3 upon PPT1 binding (Fig. 4a–c insets).

In silico approaches were used to further explore this possibility of a secondary binding site independent of a known CRD. Initial docking experiments on the modified (bound domain 3, unbound domain 5) crystal structure of domains 1–5

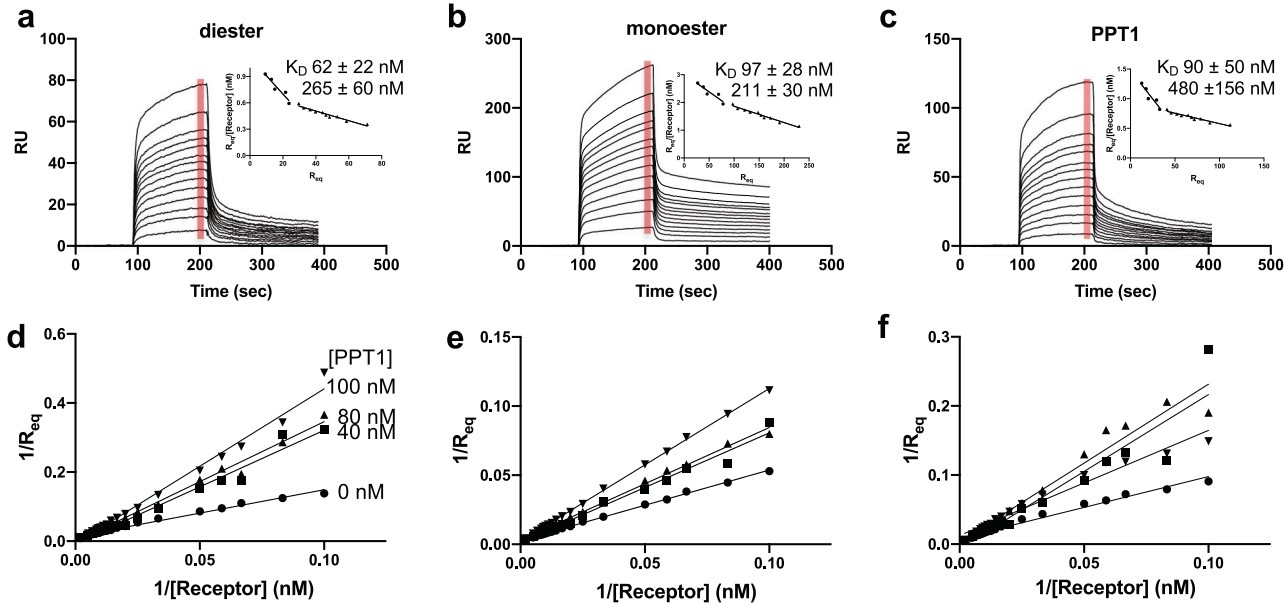

**Fig. 4 Ligand binding properties of domains 1–5 as assessed by SPR.** Sensorgrams of domains 1–5 truncated protein (10, 15, 20, 30, 40, 50, 60, 70, 80, 90, 100, 110, and 120 nM) flowing over GAA-phosphomonoester (**a**) or GAA phosphodiester (**b**) and PPT1 (**c**) surfaces. Inset graphs show Scatchard plots based on average RU value collected over 10-s time intervals at the end of the association phase for each concentration of domains 1–5 (red bar). The calculated $K_{DS}$ (−1/slope) values for two binding events are listed and $n = 4$ independent experiments (standard error of mean is reported). Results from the accompanying competitive inhibition study are displayed as double reciprocal plots (**d**–**f**) in which domains 1–5 protein (at 10, 15, 20, 30, 40, 50, 60, 70, 80, 90, 100, 110 or 120 nM concentration) was preincubated for 2 h with 0, 40, 80, or 100-nM PPT1 (as indicated in **d**) before being flowed over the three different lysosomal enzyme surfaces as indicated in **a**–**c**.

used the FTMap server to identify possible small molecule interaction sites (Fig. 6a)[39]. One of the identified "hot spots" overlaps with the species conserved region of peptide 91–101 of domain 1's β-strand 7 determined by HRPF (Supplementary Fig. 5e, f). Inspection of the X-ray structure (PDB 1EI9) revealed that PPT1's three N-glycans clustered close to each other, and molecular dynamics (MD) simulations demonstrate the inherent flexibility of glycans (Fig. 6b). Lyly et al. had previously shown that the glycan on N232 is critical for the proper trafficking of PPT1 to the lysosome[40]. Docking of the glycan on N232 into the binding site of domain 3 allows the glycan of N212 to be near the "hot spot" described above, strand 7 of domain 1, introducing a low-affinity interaction site with domain 1 (Fig. 6c). Further support of the receptor's bidentate binding arising from interactions with two individual oligosaccharides rather than with multiple arms of the same oligosaccharide comes from the work of Yamaguchi et al.[41]. This group utilized an engineered disulfide bridge containing glycopeptide comprised of two M6P-containing glycans. This peptide bound to CI-MPR with a higher affinity in the absence of reducing agent than in the presence[41]. Lysosomal enzymes typically possess multiple glycosylation sites, like PPT1. Some of these enzymes, such as β-glucuronidase, do so via oligomerization. This tetrameric protein, with four N-glycans per monomer[42], presents a more expansive spatial array of phosphorylated glycans that together can enhance the engagement with CI-MPR's CRDs to obtain high-affinity binding via multivalent interactions. The existence of low-affinity binding sites, such as that proposed above in domain 1, would enhance the avidity of these interactions.

**CI-MPR undergoes domain rearrangement at endosomal pH.** Recognition and binding of lysosomal enzymes by CI-MPR represent only the first part of this receptor's function. The receptor must also releases its cargo in the acidic pH environment

of the endosome (~pH 5). This release process is critical since the neutralization of intracellular compartments results in excessive secretion of lysosomal enzymes, with MPRs being "trapped" with their cargo[1]. Size-exclusion chromatography (SEC) of three constructs (domains 1–15, 1–5, and 7–15) shows that as the pH becomes more acidic, CI-MPR exhibits a more compact Stokes radius (Fig. 7a). To determine if changes in Stokes radius localize to one region of the receptor over another, we normalized the change in Stokes radius for each construct against the number of domains in the construct. Normalization clearly shows that the N-terminal 5 domains undergo the most substantial change in radius: they compact on each other the most (Fig. 7b). We again turned to HRPF to evaluate conformational changes of CI-MPR as a consequence of pH. This technique was shown previously to be free of pH-induced secondary effects and used to map the pH-induced structural changes of Protein G[43]. When applied to our system, analysis of HRPF results shows widespread changes in the conformation of domains 1–5 upon a change in pH from 6.5 to 4.5, with the changes more extensive than that observed upon the addition of PPT1 (Figs. 5a, 7c, and Supplementary Fig. 6). The increase in the areas of protection from oxidation upon acidification is (1) consistent with SEC data, and (2) demonstrates that CI-MPR undergoes significant domain rearrangement and compaction of its overall conformation. In addition, the two peptides (91–101 and 370–391) observed to gain protection from oxidation in the presence of PPT1 (Supplementary Fig. 5a) report higher degrees of modification in transitioning to acidic pH (Supplementary Fig. 6a), a finding consistent with ligand release and exposure of these peptides to solvent in the acidic environment of endosomes.

Concurrent with submission of this manuscript, Wang et al. published the cryo-EM structures of endogenous CI-MPR from bovine liver at both pH 4.5 and bound to IGF2 at pH 7.4[22]. However, the structure of domains 1–3 and domain 15 in the IGF2 bound cryo-EM structure at pH 7.4 was unable to be

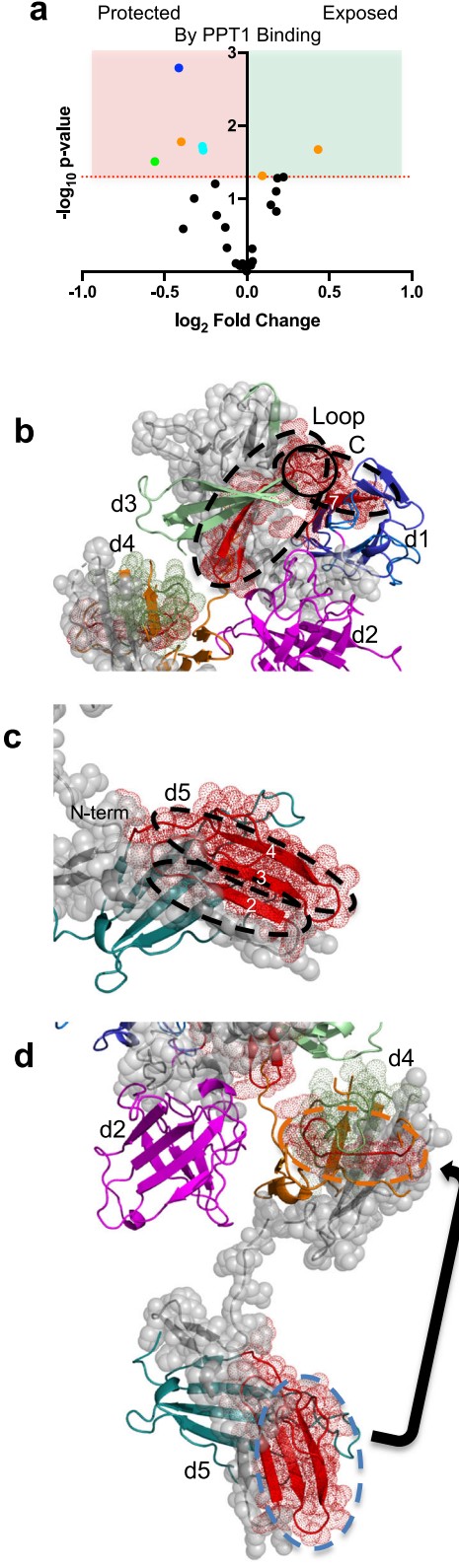

**Fig. 5 FPOP analysis of domains 1–5 in the absence and presence of PPT1 at pH 6.5. a** FPOP comparison of domains 1–5 alone and in the presence of PPT1 reveals peptides protected (red shaded region and colored by domain: d1, blue; d2, magenta; d3, green; d4, orange; or d5, cyan) or exposed (green shaded region and colored by domain) upon binding, while most peptides (black circles) show no statistically significant ($p \leq 0.05$) changes (unshaded region). **b** Comparison of peptides (dashed black ovals) (mapped onto SAXS-generated model of bound domain 3 and unbound domain 5) undergoing changes in oxidation in the absence and presence of PPT1 ligand (red mesh and ribbon, peptides more protected, green mesh and ribbon, peptides less protected in the presence of PPT1, gray spheres, no data available). Ligand binding loop C is circled. **c** Domain 5 peptides showing protection from oxidation in the presence of PPT1 mapped onto the model in **b** (dashed black ovals). β strands affected are labeled. **d** Peptides in domains 4 (orange dashed oval) and 5 (cyan dashed oval) showing changes in exposure and protection in the presence of PPT1 mapped onto the current model of domain 5 being ligand-free and domain 3 ligand bound. The arrow indicates the possible movement of domain 5, which would result in the protection of both peptides and produce an overall molecular shape consistent with SAXS data.

Domains 1 and 3, as well as domains 1 and 2, retain their association as assessed by the preservation of their domain–domain interfaces as well as minimal changes in the number of salt bridges and hydrogen bonds whether ligand is bound to domain 5 or the receptor is at low pH (Supplementary Fig. 6b). The most obvious difference between the two structures is the location of domain 4 relative to the other domains (Fig. 7d, e). It is noteworthy that the position of domain 4 at pH 4.5 is consistent with its predicted position calculated from the unbound SAXS data (Fig. 2e). In addition, when the FPOP data are mapped onto domains 1–5 of the cryo-EM structure determined at pH 4.5 (PDB 6UM1), the protection of peptides located on strands 1 and 2 of domain 4 is consistent the relocation of domain 4 (Fig. 7f). Furthermore, the position of domain 5 is very similar whether it is bound to ligand (PDB 6P8I) or at low pH (PDB 6UM1) and adopts an alternate position (based on our SAXS data), on average, at higher pH values or in the absence of ligand. This is in contrast to the other CRD of the domain 1–5 construct, domain 3, which assumes an alternate conformation when bound to ligand (Fig. 2e). Domain 3 assumes the same orientation whether unbound (PDB 1SYO) (at higher pH conditions favorable for ligand binding) or at the lower pH required for ligand release, pH 4.5 (PDB 6UM1).

## Discussion

Our X-ray crystal structures, coupled with SAXS, FPOP, and SPR analyses, allows us to propose an allosteric mechanism for the functioning of CI-MPR. Allosteric mechanisms rely on an intricate network of atomic interactions to convey binding status at one site to spatially remote sites. In the case of CI-MPR, domain 3 plays a pivotal role in this receptor's carbohydrate binding to either domain 3 or 5, facilitating a change in domain orientation, either blocking ligand accessibility or altering the stability of the binding pocket. The orientation of domain 3 of the domains 1–5 protein construct appears to be carefully regulated by a series of non-covalent interactions (Supplementary Fig. 7a–d). These loop and domain interface changes allow for the passage of information regarding ligand binding status at one CRD to another CRD through an intricate non-covalent chemical network: relaying information from one region of the molecule to a more distant region through an allosteric mechanism.

This common theme of allosteric regulation of ligand binding extends to another CI-MPR ligand: the non-M6P-containing

discerned. Comparison of domains 1–5 from the X-ray structure reported here (collected at pH 5.5 with a ligand bound to domain 5 (PDB 6P8I)) and the cryo-EM structure of endogenous CI-MPR completely ligand-unbound (PDB 6UM1) at pH 4.5, reveals an overall similarity with a calculated r.m.s.d. of 1.3 Å over 512 Cα atoms. However, overlaying the corresponding Cα atoms of domain 1 (r.m.s.d. of 0.7 Å) provides a clearer picture of the structural differences between the two pH values (Fig. 7d).

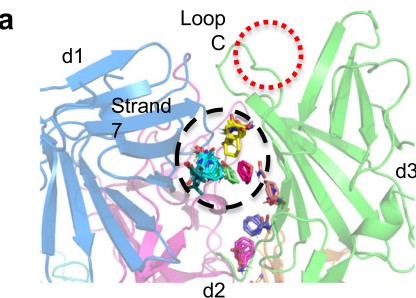

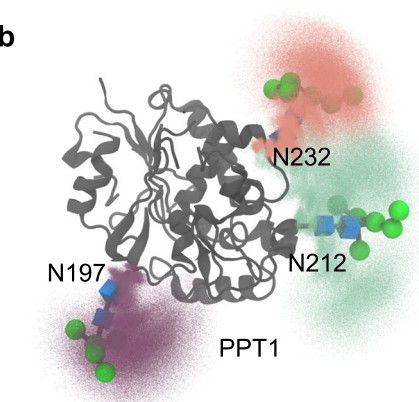

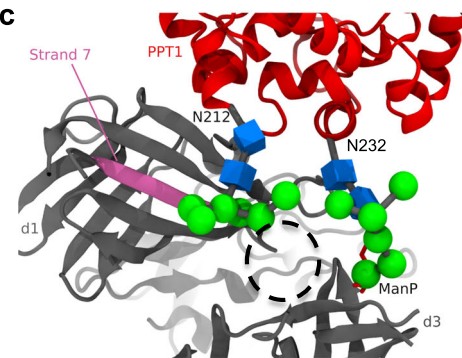

**Fig. 6 Possible secondary site of oligosaccharide interaction. a** Small molecule "hot spots" identified through the use of the FTMap server (http://ftmap.bu.edu/login.php) are shown (colored stick representation) on the model of domains 1–5 bound to a ligand in domain 3. Region of predicted small molecule interaction near secondary site proposed from analysis of FPOP data is circled in dashed black line. Loop C of the M6P binding site (circled in dashed red line) is labeled (domains are colored d1, blue; d2, magenta; d3, green). **b** Molecular dynamics simulations were used to map the extent of movement the 3 N-linked glycans of PPT1 can undergo (mannose moieties are represented by green spheres while blue boxes are used to represent GlcNAc). **c** PPT1 (PDB 1EI9) (red) structure is overlaid on the model of domains 1–5 (dark gray) with domain 3 bound and domain 5 in the unbound position with the oligosaccharide on N232 of PPT1 resting near Loop C (binding site) of domain 3. The oligosaccharide on N212 of PPT1 is located near strand 7 (magenta) of domain 1. Small molecule "hot spots" are circled in dashed black line. Mannose moieties are represented by green spheres while blue boxes are used to represent GlcNAc.

peptide, IGF2, in which contacts with domain 13 enhance the binding affinity of domain 11 for IGF2 by ~tenfold[44]. In addition, inspection of the cryo-EM structure of bovine CI-MPR bound to IGF2 suggests allosteric regulation extends to the interplay between domain 11 and domain 9. When domain 11 is bound to

IGF2, the M6P binding pocket of domain 9 is blocked and inaccessible to ligand binding (Fig. 8a). However, when the receptor is exposed to pH 4.5 conditions, domain 9 changes its relative position fully exposing its carbohydrate-binding site to solvent (Fig. 8b). These changes in the orientation of domain 9 dependent upon the binding status of domain 11 further demonstrate the allosteric behavior of this receptor. Furthermore, because domains 1–3 as well as domain 15 could not be identified in the IGF2 bound structure (PDB 6UM2) due to their high mobility[22], the effect of IGF2 binding on the other CRDs is unknown.

To our knowledge, we are reporting the first structural view of a complex between CI-MPR and a lysosomal enzyme, PPT1. This information provides insight for future studies into developing a therapy for newborns with infantile neuronal ceroid lipofuscinosis who are deficient in this enzyme. Recent studies by Amaravadi et al. showing PPT1-dependent depalmitoylation stabilize the lysosomal localization of v-ATPase subunits, which directly impacts lysosomal acidification needed for autophagic processes and mTOR signaling, further emphasizes the need for proper MPR-mediated delivery of PPT1 to the lysosome[45]. In addition, FPOP and SPR data lead us to refine our model of carbohydrate binding by domains 1–5 of CI-MPR to include the interaction of a second oligosaccharide with a neighboring domain.

To fully understand how ligand binding regulates CI-MPR structure and function, additional studies are needed to address several unanswered questions. Does lysosomal enzyme binding elicit allosteric effects on CI-MPR's non-M6P-containing ligands, IGF2, plasminogen, and uPAR? Conversely, do these non-M6P-containing ligands modify the carbohydrate-binding activity of one, several, or all four of CI-MPR's CRDs? Do the receptor's C-terminal 10 domains impact the conformational dynamics of the N-terminal five domains? Concerning the pH-dependent release of ligand, we observed construct-specific behaviors during purification over a PPT1 affinity column. Constructs containing the N-terminal domains 1–3 or 1–5, along with domains 1–15, eluted efficiently from the column upon reducing the buffer pH to 4.5, unlike the domains 7–15 construct that required M6P for elution (Supplementary Fig. 8). These findings correlate with the observation that CI-MPR's domains 1–3, but not domains 7–9 or 7–11, eluted from a pentamannosyl phosphate-agarose column at pH 4.6[46]. It is intriguing to speculate that the N-terminal 5 domains play a predominant role in regulating CI-MPR's ability to bind and release its diverse ligands. In conclusion, CI-MPR function appears to be under allosteric regulation, and future studies are needed to advance our understanding of domain interplay for this conformationally dynamic receptor.

## Methods

**Generation and expression of human CI-MPR constructs**. DNA sequences corresponding to domains 1–3 (residues 1–433), 1–5 (residues 1–726), 7–15 (residues 888–2296), and 1–15 (residues 1–2296) (numbering does not include the N-terminal signal sequence (residues 1–35) of the human CI-MPR were amplified directly from the human clone (GeneBank Accession No. J03528) obtained from the American Type Culture Collection (pGEM-8 ATCC 95661) using standard polymerase chain reaction methods. Mutant cDNAs (R391A and R688A) were generated using DpnI-mediated site-directed mutagenesis and confirmed by DNA sequencing. All constructs were cloned into pVL1392 modified to contain the native bovine CI-MPR N-terminal signal sequence followed by a *NotI* sequence and a C-terminal thrombin, hexahistidine, an Avi tag, and *XbaI* and used to generate Baculovirus using the BestBac method (Expression Systems). *Spodoptera frugiperda* (Sf9) cells (Expression Systems) were infected with baculovirus at a density of $3.0 \times 10^6$ cells per ml for 96 h at 27 °C in ESF 991 with 1% Production Boost Additive added after 24 h. Cells were removed from the medium by centrifugation at 1000 *g*.

**Purification of human CI-MPR domains 1–5 protein for PDB 6P8I**. The media was concentrated to ~40 ml using Amicon stir cells before dialysis against three

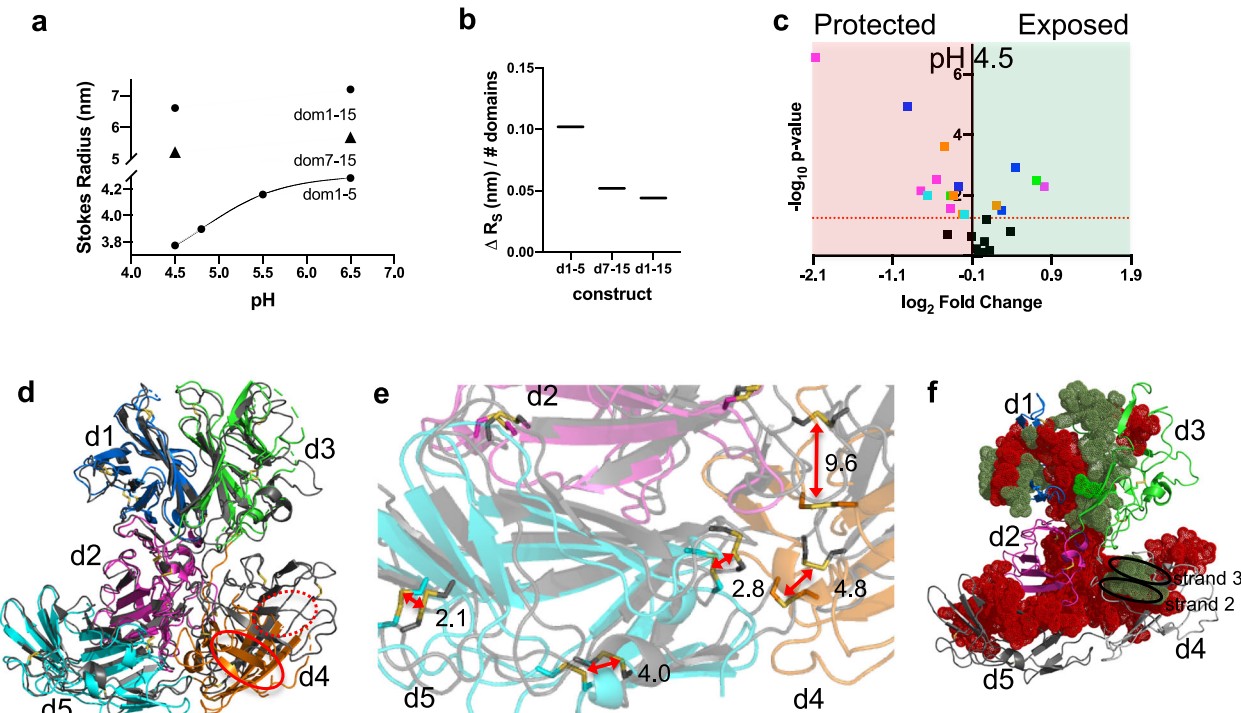

**Fig. 7 Domains 1–5 adopt a more compact conformation at pH 4.5 compared to pH 6.5. a** A plot of the calculated Stokes radius derived from SEC data collected under different pH conditions for domains 1–5, domains 7–15, and domains 1–15. **b** The observed changes in calculated Stokes radius ($\Delta R_s$) for domains 1–5, 7–15, and 1–15 normalized per number of domains in each construct. **c** FPOP analysis of domains 1–5 at pH 6.5 versus 4.5 revealing peptides protected (red shaded region) or exposed (green shaded region) (colored by domain: d1, blue; d2, magenta; d3, green; d4, orange; or d5, cyan) upon lowering of pH to that of the endosome. **d** The overlay of Cα atoms of domain 1 of N-terminal 5 domains of full-length CI-MPR of the cryo-EM structure determined at pH 4.5 (gray ribbon) (PDB 6UM1) with our domain 1–5 structure crystallized at pH 5.5 (PDB 6P8I) (domain 5 bound to the oligosaccharide of a crystallographic neighbor). Strands 2 and 3 of both domain 4 structures have been labeled and circled with either a solid red line (PDB 6P8I) or a dashed red line (6UM1) to illustrate displacement of domain 4. **e** Enlargement of the area in **d** showing the intersection of domains 2, 4, and 5. Displacement of sulfur atoms (reported in Å) of disulfide bonds (red arrows) between structures PDB 6UM1 and PDB 6P8I illustrating the change in orientation of domains at lower pH and in the absence of a bound CRD. **f** Mapping of FPOP data from **c** and Supplementary Fig. 6a onto domains 1–5 of pH 4.5 cryo-EM model (PDB 6UM1). Peptides showing a higher degree of protection at pH 4.5 versus 6.5 are mapped (red spheres) onto the SAXS-based model of domains 1–5 in the absence of ligands, while those showing less protection are mapped as green spheres. Model regions undergoing no statistically significant changes or lacking data are represented as ribbons. The lowering of the pH causes strand 2 of domain 4 to become less protected while the atoms of strand 3 (circled in black) become more protected under these conditions.

times 4 L of 20-mM Tris, pH 7.5 at 22 °C, 150-mM NaCl. Protein solutions were centrifuged at 20,000 g to remove particulate matter before loading on 5-ml Ni-NTA columns. The resin was washed with 20-mM Tris pH 7.6, 300-mM NaCl, and 20-mM imidazole before elution with 20-mM Tris pH 7.6, 300-mM NaCl, and 100-mM imidazole. Fractions were analyzed by SDS-Page, pooled, and concentrated to 1 mg/ml before overnight dialysis at 4 °C into 20-mM Tris, pH 7.6 at 22 °C, 150-mM NaCl. C-terminal tags were removed by incubation overnight at 4 °C with Thrombin (Sigma). Thrombin was removed by passage over benzamidine agarose beads. Protein was incubated overnight with PNGaseF to remove N-linked glycans followed by passage over Ni-NTA agarose to remove His-tagged PNGaseF. Domains 1–5 protein was then passed over a 10/300 Superdex G200 column equilibrated in 20-mM Tris pH 7.4 at 22 °C, 150-mM NaCl to remove any remaining aggregates or contaminants.

**Crystallographic studies for PDB 6P8I.** Protein was concentrated to 7 mg/ml and incubated with 10-mM M6P and 10-mM MnCl₂. Initial crystallization hits were found with Molecular dimensions JCSG-plus. Optimization was carried out using the hanging drop method of vapor diffusion in 24-well Falcon plates with 500 μl of well solution (100-mM ammonium citrate dibasic, pH 5.5, 20% PEG 3350) at 19 °C. Protein solution was dispensed on cover slips in a 1:1 ratio with well solution. Crystals appeared in greater than a week. Crystals were cryoprotected in reservoir solution + 25% glycerol and frozen in liquid nitrogen. Data were collected at 1.54 Å using a Rigaku M007 generator equipped with Osmic mirrors and an R-AXIS IV++ detector. Data were processed and scaled with HKL2000 software[47].

**Purification of human CI-MPR domains 1–5 for PDB 6V02.** Protein was prepared as described above except without PNGaseF digestion and additional purification over a 5-ml pentamannosyl phosphate-agarose affinity column[19]

equilibrated in 50-mM imidazole, pH 6.5, 150-mM NaCl, 5 mM β-glycerol phosphate. Protein was eluted by the addition of 10-mM M6P to column buffer.

**Crystallization studies for PDB 6V02.** Protein was concentrated to 7 mg/ml and incubated with 10-mM MnCl₂. A crystallization hit was identified using Molecular dimensions JCSG-plus (100-mM HEPES, 30% Jeffamine ED-2003) after more than 1 week at 19 °C (sitting drops with 1:1 ratio of protein (0.15 μl) to well solution (60 μl in well). Crystals were cryoprotected in reservoir solution + 20% glycerol and frozen in liquid nitrogen. Data were collected at 1 Å at APS beamline 17-ID (IMCA-CAT) at 100° K, processed using autoPROC[48].

**Structure determination of human CI-MPR domains 1–5.** Phases for the pH 5.5 conditions structure were determined by Phaser in CCP4i[49] using homology models generated using the Swiss-Model server. The model was iteratively refined using PHENIX[50] and manually rebuild in COOT[51] and showed reasonable stereochemistry, with 85.5% in the Ramachandran favored zones, 11.3% in the allowed, and 3.2% outliers. The pH 7.0 structure was also solved by molecular replacement using PHASER and the domain 1–5 structure previously solved. The model was refined using PHENIX and iteratively rebuild in COOT and showed reasonable stereochemistry, with 83.2% in the Ramachandran favored zones, 12.6% in the allowed, and 4.2% outliers.

**Expression and purification of PPT1.** Recombinant human PPT1 protein was expressed and purified following the protocol outlined in Lu et al. 2010[52].

**Generation of model from PDB 68PI with M6P bound to domain 3.** We used SwissModel to generate a model of human domains 1–5 bound to M6P based on

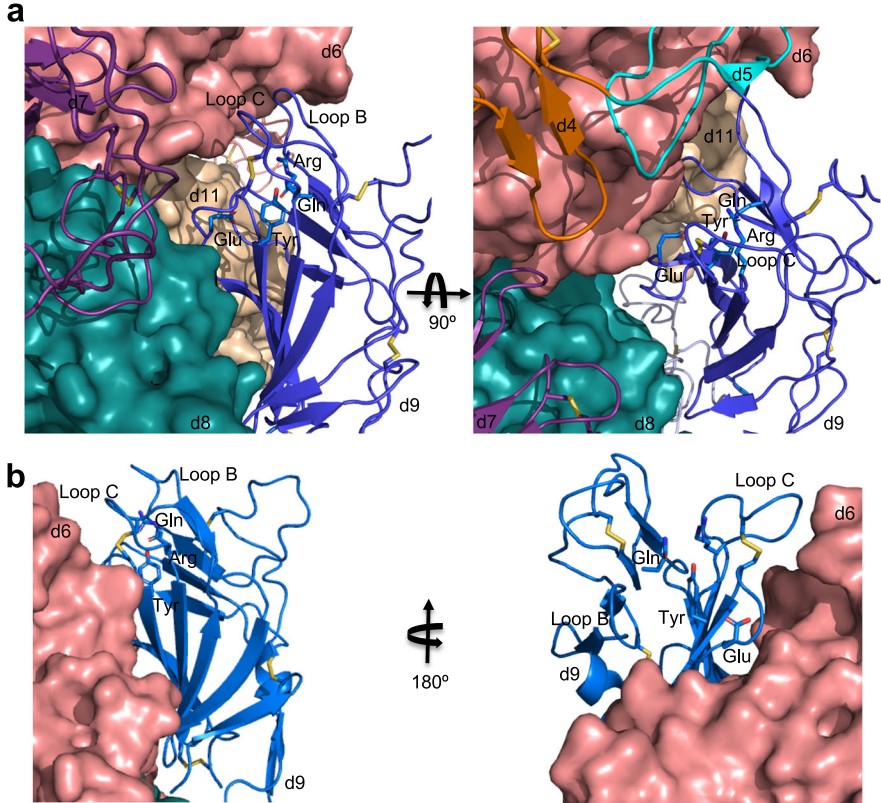

**Fig. 8 Occlusion of carbohydrate-binding site of domain 9 of endogenous bovine CI-MPR in the presence of IGF2 (PDB 6UM2) supporting hypothesis that CI-MPR ligand binding is allosterically regulated. a** Cartoon representation showing domain 9's (blue) C-terminal β sheet's interaction with the N-terminal β sheet of domain 8 (teal). The binding site of domain 9 faces into domain 6 (salmon) restraining loops B and C essential for ligand binding. The four essential carbohydrate-binding residues (sticks: Gln, Arg, Glu, and Arg) of domain 9 are shown. Adjacent domains (domains 6, 8, and 11) are represented as molecular surfaces. The second panel is rotated 90° downward along the *x*-axis relative to the first panel. **b** Cartoon representation illustrating the change of position in domain 9 of CI-MPR (PDB 6UM1) when exposed to pH 4.5 buffer conditions compared to pH 7.4 in the presence of IGF2 (**a**). For clarity, the second panel is rotated 180° along the *y*-axis relative to the original showing the change in solvent accessibility of the binding site at pH 4.5.

our two structures: PDB 1SYO and PDB 6P8I[53]. We next superimposed the Cα atoms of domain 1 of the two structures and then appended the transposed coordinates for domains 1–4 to the bound human domains 1–3 SwissModel generated model yielding a new model of human domains 1–5 with ligand bound to domains 1 and 5.

**Purification of protein for other biophysical studies**. Human domain 1–5 protein for SAXS, SPR, negative stain, EM, and SEC was generated as follows. Cells were removed from the medium by centrifugation at 1000 *g* and the pH of the medium was adjusted to pH 8.0 with 10-M NaOH. Precipitates were removed from the medium by centrifugation at 4000 *g* before loading over 5 ml of PROTEINDEX Ni-Penta agarose 6 fast flow resin (Marvelgent Biosciences). The resin was washed with 20-mM Tris pH 7.6, 300-mM NaCl, and 20-mM imidazole before elution with 20-mM Tris pH 7.6, 300-mM NaCl, and 100-mM imidazole. Fractions were analyzed by SDS-PAGE, pooled, and concentrated to 1 mg/ml before overnight dialysis at 4 °C into 20-mM Tris, pH 7.6 at 22 °C, 150-mM NaCl. C-terminal tags were removed by incubation overnight at 4 °C with Thrombin (Sigma). Thrombin was removed by passage over benzamidine agarose beads. Domain 1–5 protein was also incubated with PNGaseF overnight at 22 °C to remove N-linked glycans. Proteins were passed over Ni-NTA agarose to remove cleaved His-tags as well as PNGaseF (His-tagged). Flow-through and proteins eluted with 20-mM Tris pH 7.5 at 22 °C, 150-mM NaCl were combined and dialyzed overnight at 22 °C in column buffer (50-mM imidazole, pH 6.5, 150-mM NaCl, 5 m β-glycerol phosphate, 10-mM MnCl₂). Protein was further purified by passing dialyzed proteins over a 1-ml PPT1 amine coupled agarose resin column (10 mg/ml) (Pierce NHS-activated agarose slurry). Loaded protein was washed with column buffer before elution with 20-mM MES, pH 4.5, 150-mM NaCl, 5 mM β-glycerol phosphate, 10-mM MnCl₂ and neutralization with Tris buffer to pH 7.4. Any aggregates or contaminants were removed by passage over a Superdex G200 10/300 column equilibrated in column buffer without MnCl₂. Monomeric fractions were pooled and stored at 4 °C. Protein concentration was determined using the Bradford assay (Bio-Rad) with bovine serum albumin as the standard.

**Purification of other CI-MPR constructs**. Domains 1–3, 7–15, and 1–15 protein for SPR and/or SEC were expressed in Sf9 cells grown, harvested, and purified as described in the previous section without treatment with thrombin and PNGaseF.

**LC-MS/MS analysis of PPT1 glycosylation**. Aliquots of PPT1 (20 μg) were reduced, carboxyamidomethylated, dialyzed against nanopure water at 4 °C overnight, and then dried in a Speed Vac. The dried, desalted sample was resuspended and digested with trypsin (Promega, sequence grade) at 37 °C overnight. Following digestion, the sample was again dried and subsequently resuspended in solvent A (0.1% formic acid in water) and passed through a 0.2-μm filter (Nanosep, PALL) before analysis by LC-MS/MS.

LC-MS/MS analysis was performed on an Orbitrap Fusion equipped with an EASY nanospray source and Ultimate 3000 autosampler LC system (Thermo Fisher). Resuspended tryptic peptides were chromatographed on a nano-C18 column (Acclaim pepMap RSLC, 75 μm × 150 mm, C18, 2 μm) with an 80-min gradient of increasing mobile phase B (80% acetonitrile, 0.1% formic acid in distilled H₂O) at a flow rate of 300 nl/min routed directly into the mass spectrometer. Full MS spectra were collected at 60,000 resolution in FT mode and MS/MS spectra were obtained for each precursor ion by data-dependent scans (top-speed scan, 3 s) utilizing CID, HCD, or ETD activation and subsequent detection in FT mode.

Phosphorylated glycopeptides were annotated by manual data interpretation of the LC-MS/MS data following initial processing by Byonic software (Protein Metrics). Byonic parameters were set to allow 20 ppm of precursor ion monoisotopic mass tolerance and 20 ppm of fragment ion tolerance. Byonic searches were performed against the human PPT1 sequence allowing modification with phosphorylated and non-phosphorylated human/mammalian N-glycans.

**SPR studies of CI-MPR domains 1–5**. All SPR measurements were performed at 25 °C using a Biacore 3000 instrument (BIAcore, GE Healthcare, Piscataway, NJ) as previously described[17,18]. Purified lysosomal enzymes (GAA mono- and diester, PPT1) were immobilized at a density of ~1000 RU on a CM5 sensor chip by

primary amine coupling following the manufacturer's procedure. The reference surface was treated the same way except for no protein addition. Purified domains 1–5 and domains 1–5 with PPT1 were prepared in 50-mM imidazole, 150-mM NaCl, 5-mM MgCl₂, 5-mM MnCl₂, 5-mM CaCl₂, pH 6.5 supplemented with 0.005% (v/v) P20. All samples were incubated for 2 h before loading on the instrument. Samples were injected in a volume of 80 μl over the reference and coupled flow cells at a flow rate of 40 μl/min for 2 min before dissociation with buffer alone for 2 min. The sensor chip surfaces were regenerated with a 20-μl injection of 10-mM HCl at a flow rate of 10 μl/min and allowed to re-equilibrate with running buffer for 1 min before the next injection. The response at equilibrium ($R_{eq}$) was determined for each concentration of protein/complex by averaging the response over a 10-s period within the steady-state region of the sensorgram (BIAevaluation software package, 4.0.1). Scatchard analysis was performed to determine linear regions (10–40 nM) and (50–200 nM). The $R_{eq}$ was plotted for these two regions versus the concentration of protein and fit to a 1:1 binding isotherm. All responses were double-referenced by subtracting the change in refractive index for the flow cell derivatized in the absence of protein from the binding sensorgrams.

**SAXS data collection on CI-MPR domains 1–5.** SAXS was performed at BioCAT (Sector 18) Advanced Photon Source utilizing a Pilatus 1M detector. Data was collected at ~20 °C with a wavelength of 1.033 Å and ~3.5 m sample-to-detector distance ($q$ range = 0.00535–0.387 Å⁻¹). Before introduction into the stationary SAXS quartz capillary (1.5 mm ID, 1.52 mm OD), 0.5 mg of domains 1–5 protein was incubated with 10-mM M6P in 50-mM imidazole, pH 6.5, 150-mM NaCl, 5-mM β-glycerol phosphate for 1 h at 22 °C. Batch mode SAXS data were collected on human domains 1–5 alone, and in complex with M6P. SEC-SAXS was performed on domains 1–5 in complex with PPT1, PPT1 alone and domains 1–5 protein alone. For the complex, 1.25 mg of domains 1–5 was incubated with 5 mg of PPT1 in the above imidazole buffer for 1 h at 22 °C before data collection. SEC/SAXS data were collected simultaneously (0.5-s exposures collected every 3 s) upon elution from a 10/300 Superdex G200-Increase column equilibrated in matched buffer and at a flow rate of 0.75 ml/min. Free domains 1–5 protein could not be completely separated from that in complex with PPT1 as seen in the chromatogram. Exposures flanking the elution peaks were averaged to generate the $I(q)$ versus $q$ curve for the buffer and then subtracted from the elution peak curves to obtain the sample SAXS curves. Data were processed with Primus[54] and ab initio dummy atom modeling was done with DAMMIF[55]. The merged SAXS curves were used to generate pair distribution functions, $P(r)$, and Kratky plots (PRIMUS). The flexibility analysis curves were generated using SCATTER 3.0 software. The FoXS server was used to compute the SAXS profile using the coordinates from the pH 5.5 structure (PDB 6P8I). The MultiFoXS server was used to calculate the population-weighted ensemble fitting to the unbound protein scattering curves.

**EM on CI-MPR domains 1–5.** The negative-stained EM specimens of domains 1–5 and the domains 1–5 bound to M6P were prepared as described above. In brief, the samples were diluted to ~0.001 μg mL⁻¹ with sample buffer. An aliquot (~4 μL) of diluted sample was placed on an ultra-thin carbon-coated 200-mesh copper grid (CF200-Cu-UL, Electron Microscopy Sciences, Hatfield, PA, USA, and Cu-200CN, Pacific Grid-Tech, San Francisco, CA, USA) that had been glow-discharged for 15 s. After 1-min incubation, the excess solution on the grid was blotted with filter paper. The grid was then washed with water and stained with 1% (w/v) uranyl formate before air-drying with nitrogen. The EM samples were examined by using a Zeiss Libra 120 Plus TEM (Carl Zeiss NTS) operated at 120-kV high tension with a 10–20 eV energy filter. The OpNS micrographs were acquired under defocus at ~0.6 μm and a dose of ~40–90 e⁻·Å⁻² using a Gatan UltraScan 4K × 4K CCD under a magnification of 80 kx (each pixel of the micrographs corresponds to 1.48 Å in specimens). The contrast transfer function of each micrograph was examined by using *ctffind3* software[56] and the phase and amplitude were corrected by using the "TF CTS" command in SPIDER[57] software or GCTF[58] after the X-ray speckles were removed. Particles were then selected from the micrographs by using *boxer* (EMAN software[59]). All particles were masked by using a round mask generated from SPIDER software after a Gaussian high-pass filtering. The 50 reference-free class averages of particles were obtained by using *refine2d* (EMAN software) based on ~3000 particles windowed from ~140 micrographs.

**Size-exclusion chromatography of truncated constructs.** A G200-Increase 10/300 column was run at a flow rate of 0.75 ml/min and equilibrated with either pH 6.5 buffer (20-mM imidazole, 150-mM NaCl, pH 7.5), pH 5.5 buffer (20-mM sodium citrate, 150-mM NaCl, pH 5.5), pH 4.8 buffer (20-mM sodium citrate, 150-mM NaCl, pH 4.8), or pH 4.5 buffer (20-mM sodium citrate, 150-mM NaCl, pH 4.5). Domains 1–5 protein was run at all the above-listed pH values while domains 7–15 and 1–15 were run at pH 6.5 and 4.5. All proteins were injected onto the column as 50 μg in 200 μl of matched pH buffer. Stokes radius was calculated using thyroglobulin (bovine thyroid), β-amylase (sweet potato), albumin (bovine albumin), carbonic anhydrase (erythrocytes), and cytochrome c (horse heart).

**FPOP of human CI-MPR domains 1–5.** A final concentration of 5-μM domains 1–5 protein was incubated in the 5-mM sodium citrate buffer in the presence or absence of 5-μM PPT1 at pH 6.5 for 1 h. For FPOP at pH 4.5, 5-mM sodium citrate buffer was used to incubate CI-MPR domain 1–5 for 1 h without PPT1. FPOP was performed as described previously[60]. Briefly, 20 μl of protein sample mixture containing 1-mM adenine, 17-mM glutamine, and 100-mM hydrogen peroxide was irradiated by flow through the path of the pulsed ultraviolet laser beam from a Compex Pro 102 KrF excimer laser (Coherent, Germany). The laser fluence was calculated to be ~10.1 mJ/mm²/pulse. The laser repetition rate was 15 Hz. The flow rate was adjusted to 13 μL/min to ensure a 15% exclusion volume between irradiated segments. After laser illumination, each replicate was collected in a microcentrifuge tube containing 25 μl of quench mixture that contained 0.5-μg/μl H-Met-NH₂ and 0.5-μg/μl catalase to eliminate secondary oxidants. The adenine hydroxyl radical dosimetry readings were measured at 265 nm in nanodrop (Thermo Scientific) to ensure that all the samples were exposed to equivalent amounts of hydroxyl radical[61]. All FPOP experiments were performed in triplicate for statistical analysis.

After FPOP and quenching, 50-mM Tris, pH 8.0 containing 1-mM CaCl₂, 5-mM DTT was added to the protein samples and incubated at 95 °C for 15 min to denature and reduce the protein. The sample was cooled on ice, trypsin with 1:20 ratio of trypsin:protein was added and incubated at 37 °C for 12 h with rotation. Sample digestion was stopped by adding 0.1% formic acid and the samples were analyzed on a Dionex Ultimate 3000 nano-LC system coupled to an Orbitrap Fusion Thermo Scientific (San Jose, CA). Samples were trapped on a 300-μM id X5 mm PepMap 100, 5-μm (Thermo Scientific) C18 trapping cartridge, then back-eluted onto an Acclaim PepMap 100 C18 nanocolumn (0.75 × 150 mm, 2 μm, Thermo Scientific). Separation of peptides on the chromatographic system was performed using a binary gradient of solvent A (0.1% formic acid in water) and solvent B (0.1% formic acid in acetonitrile) at a flow rate of 0.30 μL/min. The peptides were eluted with a gradient consisting of 2–10% solvent B over 4 min, increasing to 32% B over 25 min, ramped to 95% B over 4 min, held for 4 min, and then returned to 2% B over 2 min and held for 8 min. Peptides were eluted directly into the nanospray source of an Orbitrap Fusion instrument using a conductive nanospray emitter obtained from Thermo Scientific. All the data were collected in positive ion mode. Collision-induced dissociation CID was used to fragment peptides, with an isolation width of 3 *m/z* units. The spray voltage was set to 2400 V, and the temperature of the heated capillary was set to 300 °C. Full MS scans were acquired from *m/z* 350 to 2000 followed by eight subsequent MS2 CID scans on the top eight most abundant peptide ions.

Peptides from tryptic digests of CI-MPR domain 1–5 were identified using ByOnic version v2.10.5 (Protein Metrics). The search parameters included all possible major oxidation modifications as variable modifications and the enzyme specificity was set to cleave the protein after arginine and lysine residues. The peak intensities of the unoxidized peptides and their corresponding oxidation products observed in LC-MS were used to calculate the average oxidation events per peptide in the sample as previously reported[37]. Briefly, peptide level oxidation was calculated by adding the ion intensities of all the oxidized peptides multiplied by the number of oxidation events required for the mass shift (e.g., one event for +16, two events for +32) and then divided by the sum of the ion intensities of all unoxidized and oxidized peptide masses as represented by Eq. (1)

$$P = [I(+16) \text{oxidized} \times 1 + I(+32) \text{oxidized} \times 2 + I(+48) \text{oxidized} \times 3 + \dots / \\ [I \text{unoxidized} + I(+16) \text{oxidized} + I(+32) \text{oxidized} + I(+48) \text{oxidized} \dots], \quad (1)$$

where $P$ denotes the oxidation events at the peptide level and $I$ values are the peak intensities of oxidized and unoxidized peptides[61].

**Generation/energy minimization of a glycosylated PPT1 model.** The 3D structure of PPT1 (PDB code 3GRO) with M6GN2 (DManpα1–6[DManpα1–3] DManpα1–6[DManpα1–2DManpα1–3]DManpα1–4DGlcpNAcβ1–4DGlcpNAcβ1-) conjugated to (N197, N212, and N232) using the glycoprotein builder available at GLYCAM-Web (www.glycam-web.org) and an in-house program that adjusts the glycosidic linkages to relieve any atomic overlaps between the conjugated glycan and the underlying protein. The glycosylated PPT1 structure was placed in a periodic box of ~15,000 TIP5P waters with a 10 Å buffer between the glycoprotein and the box edge. Energy minimization of all atoms was performed for 20,000 steps (10,000 steepest descent, followed by 10,000 conjugant gradient).

**MD simulations.** All MD simulations were performed with the CUDA implementation of the PMEMD[62] simulation code, as present in the Amber14 software suite[63]. The GLYCAM06 force field[64] and Amber14SB force field[65] were employed for the carbohydrate and protein moieties, respectively. A Berendsen barostat with a time constant of 1 ps was employed for pressure regulation, while a Langevin thermostat with a collision frequency of 2 ps⁻¹ was employed for temperature regulation. A nonbonded interaction cutoff of 8 Å was employed. Long-range electrostatics were treated with the particle-mesh Ewald method[66]. Covalent bonds involving hydrogen were constrained with the SHAKE algorithm, allowing an integration time step of 2 fs[67] to be employed. All simulations were performed

under nPT conditions and the restraints employed were 5-kcal/mol-Å² Cartesian. The energy minimized coordinates were equilibrated at 300 K over 400 ps with restraints on the solute heavy atoms. The system was then equilibrated with restraints on the Cα atoms of the protein for 1 ns, before performing a production MD simulation for 500 ns, but with restraints applied only to the Cα atoms of residues on either sides of gaps, namely (11, 12, 20, 21, 25, 26, 44, 45, 78, 79, 98, 99, 184, 185, 190, 191, 236 and 237. AMBER numbering).

**Modeling of interaction between PPT1 and domains 1–5**. The glycosylated PPT1 structure was aligned to the Man6P in the d3 binding site of the "bound" form of the d1–5 structure. The alignment was performed by superimposing the nonreducing terminal mannose residue on the DManpα1–2DManpα1–3 arm of M6GN2 at N197 onto the complexed Man6P. This process was repeated for 100 snapshots taken at regular intervals from the MD simulation trajectory. An in-house program was employed to adjust the glycosidic linkages in N232 and N212 to bring N212 into contact with the region identified by the "hot spot" analysis. The program adjusts the glycosidic linkages within known low-energy ranges[68] while avoiding atomic overlaps, as described previously[69].

**Fitting model of domains 1–5 bound to PPT1 to SAX data**. The UCSF-Chimera 1.12[70] program was employed to fit the co-complex of d1–5 with glycosylated PPT1. Snapshots from the MD simulation of PPT1 were aligned to the "bound" form of d1–5 via N232 as described in the previous "Modeling of multimeric interaction" section. These structures were then fit into the SAX data using the map fitting feature of UCSF Chimera with 50 fittings per structure and a 90% cutoff.

**Statistics and reproducibility**. Statistical analysis was performed using Prism software for SPR experiments and EXCEL for FPOP.

**Reporting summary**. Further information on research design is available in the Nature Research Reporting Summary linked to this article.

## Data availability

All data generated or analyzed during this study are included in the published article and its Supplementary Information. The X-ray crystal structures and structure factors of human domains 1–5 of CI-MPR at pH 5.5 and 7.0 have been deposited in Protein Data Bank under accession codes PDB 6P8I and PDB 6V02. SAXS data have been deposited at SASDBD with codes: SASDHL4 (N-terminal domains 1–5 of the cation-independent mannose 6-phosphate receptor (CI-MPR)), SASDHM4 (N-terminal domains 1–5 of the cation-independent mannose 6-phosphate receptor (CI-MPR) from SEC-SAXS), SASDHN4 (N-terminal 5 domains of the cation-independent mannose-6-phosphate receptor (CI-MPR) bound to mannose 6-phosphate (M6P)), SASDHP4 (palmitoyl-protein thioesterase 1 (PPT1)), and SASDQ4 (N-terminal domains 1–5 of the cation-independent mannose 6-phosphate receptor (CI-MPR) in complex with palmitoyl-protein thioesterase 1 (PPT1)).

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

## Acknowledgements

We would like to thank Dr Jui-Yun Lu and Dr Sandra L. Hofmann for providing the CHO cell line expressing recombinant PPT1. We would like to thank Dr Srinivas Chakravarthy, BioCAT, Argonne National Laboratory, sector 18 for SAXS data collection and processing during their fall 2017 SAXS data collection workshop, which was supported by grant to Thomas Irving 9 P41 GM103622 from the National Institute of General Medical Sciences of the National Institutes of Health (NIH). We would like to thank Richard Bohnsack for generating the human CI-MPR domains 1–5 clone and Dr Lei Zhang's early screening of the sample by negative-staining TEM. This research used resources of the Advanced Photon Source, a US Department of Energy (DOE) Office of Science User Facility operated for the DOE Office of Science by Argonne National Laboratory under Contract No. DE-AC02–06CH11357. Work at the Molecular Foundry was supported by the Office of Science, Office of Basic Energy Sciences, of the US Department of Energy under Contract No. DE-AC02–05CH11231. G.R. acknowledges support from the National Health Institute, NIGMS (RO1HL115153). S.K.M. and J.S.S. acknowledge support from the National Institute of General Medical Sciences (P41GM103390 and RO1GM127267). R.J.W. thanks the National Institutes of Health (U01 CA 207824 and P41 GM103390) for financial support. This work was supported by the National Institute of Diabetes and Digestive and Kidney Diseases of the NIH under award number R01DK042667 to N. M.D. and supported in part by Institutional Research Grant #16-183-31 from the American Cancer Society and the MCW Cancer Center through a pilot research grant to L.J.O. The content is solely the responsibility of the authors and does not necessarily represent the official views of the NIH.

## Author contributions

N.M.D. and L.J.O. initiated the project. L.J.O. performed mutagenesis, protein expression, purification, crystallization, X-ray data collection (PDB 6P8I) and processing (PDB 6P8I), determined structures and structural analysis (PDB 6P8I and PDB 6V02). K.P.B. collected and processed the data for the pH 7.0 structure (PDB 6V02). L.J.O. conducted SPR experiments and processed/analyzed SPR data, analyzed SAXS data, conducted and analyzed SEC data. G.R. analyzed and supervised the collection of negative stain and EM data. S.K.M. collected and analyzed the FPOP data under the supervision of J.S.S. L.J.O. and A.S. ran FTmap. Modeling done by O.G. under the supervision of R.J.W. Mass spectrometry done by M.I. and supervised by M.T. All authors contributed to data interpretation and preparation of the manuscript. Initial manuscript written by LJO with later versions edited by N.M.D. and J-J.P.K. N.M.D., J-J.P.K., and L.J.O. orchestrated the project.

## Competing interests

The authors declare no competing interests.
