## [Peer Review File · Communications Biology]

Reviewers' comments:

Reviewer #1 (Remarks to the Author):

This paper describes a complex series of structural, SPR binding, FPOP, and modeling studies on a multi-domain fragment of the cation independent mannose-6-phosphate receptor (CI-MPR) to examine domain-specific conformational changes that occur upon ligand binding and pH dependent ligand release. It is a complicated manuscript that extends prior data from the same lab on the structural basis of ligand interactions for the CI-MPR, a multidomain receptor that binds M6P-containing ligands using several of its MRH domains as well as interactions with other protein ligands with other domains. The authors have previously determined crystal structures of the bovine CI-MPR for domains 1-3 as well as domain 5 and others have determined structures of domains 11-14. Domains 3, 5, 9, and 15 have been shown to bind M6P-containing ligands. The present studies focus on crystal structures of domains 1-5 at pH 5.5 and 7.0. The authors then compare these structures with the prior domain 1-3 structures and perform SAXS, SPR, negative stain EM, and FPOP analysis at different pHs and in the presence of glycosylated ligands, and modeling of domain interactions.

While the manuscript is generally well-written, and the structural, FPOP, SPR, and modeling data are superb, there are several concerns regarding the data presentation that make the manuscript more confusing than necessary. The biggest challenge of the manuscript is the somewhat disconnected collection of data that is brought together to describe the nature of the domain interactions and conformational changes that occur during ligand binding and pH dependent binding and release of ligands. The manuscript starts with the presentation of X-ray structure data for domains 1-5 at 2 different pHs (pH 5.5 and 7) that show essentially the same protein structure at both pHs. This is a rather surprising result, since it is clear based on prior data from the CI-MPR and essentially all of the other data in the present manuscript that there are major conformational changes that occur during ligand binding and release for the domain 1-5 segments of the protein. The remainder of the manuscript is then focused on collecting data that indicate that the compressed domain assembly identified in the crystal structure does not reflect the dynamics of domain interactions of the protein in solution. The dichotomy between the identical, static X-ray structures at pH 5.5 and 7 versus significant structural differences in conformational dynamics in solution lead to a model for allosteric regulation of ligand interactions through global conformational changes in domain positioning.

Probably the largest concerns in the manuscript are the relatively brief and non-descriptive figure legends that make interpretation of the figures and data a challenge. Fig. 1 and its legend start with some of these concerns. Panel 1B supposedly shows an overlay of the pH 5.5 and pH 7 structures, but it is almost impossible to even discern that there is even a grey cartoon representation for the pH 7 structure in the figure. Panel 1B has domain labels that are either hidden behind the protein structure or are far away from the respective domain. In most of the other panels the domains are labeled d1, d2, d3, etc, but in Panel 1B they are labeled 1, 2, 3, etc. The dotted circle in Panel 1B is also described as a star in the legend. The glycan associated with d5 in Panel 1B is also so light that it is difficult to see. In Panel 1C it is not clear from the figure legend why they are even showing the 1SYO structure. It is also not clear until Panel 1H that the structures in Panel 1B and Panel 1C are essentially identical for d1 and d2. Are they supposed to be aligned based on those domains? If not, they really should be. The M6P bound in Panel 1C appears to be a stick representation, but why are there stick representations of a glycan attached to d1? Is that a covalently attached glycan? It is not labeled in the figure nor described in the legend. The differences in position and angle of d3 in Panel 1D are also hard to interpret since there is no frame of reference of an anchor position based on the positions of d1 or d2 in the panel to interpret the $\sim 45^\circ$ angle. The 45° label is also positioned partially behind the protein structure. Overall, Panels 1B-D are poorly described in the legend for a reader to take home the information content that the author intended.

Similarly, the comparisons in Panels 1E and 1F are confusing since the authors do not use a similarly positioned domain d1 and d2 in the two panels. These two domains are the frame of reference for the change in position in d3. They should be oriented the same or the point of the figure is lost. Fig. 1H is wonderful in its comparison of the position of domain d3 and should have been used for Panels 1B and 1C, except that the poor contrast in colors for domains d3 bound and d3 unbound (light green and not quite so light green) make it very difficult to tell that there are even two different d3s in the panel.

Panels 1I and 1J are an interesting approach for presentation for the domain interfaces, but why is the full surface of domain d3 shown in Panel 1I when only the interfaces are shown for the other domains in the panels? Overall, the entire figure legend needs to be a more expansive in description for what the authors want the reader to see. This expanded description is not present in either the legend or in the text of the Results section and the reader is left trying to interpret the panels in the figure without sufficient information.

For Fig. 2 and Supplementary Fig 3 it is clear that the initial crystallographic models do not fit the SAXS envelopes well (measures of fit should be included for all panels in Fig. 2 and S3) and the extension of domain d5 appears to improve the fit indicating that there is likely significant dynamics of the respective domains in solution. However, the take home message of Supplementary Fig. 3 was hampered by the lack of a legend for Panel E, where the extension of domain d5 allowed an improved fit into the SAXS envelope. This hypothesis was supported by the negative stain EM imaging (Fig. 3), but again more expansive figure legends with potential interpretations of the figure data would help the reader understand what they were looking for. The lack of detail and interpretation in the legends and/or the Results section makes it hard to digest the data in those figures.

The data in Fig. 4 and Supplementary Fig 4 using SPR and competitive SPR were quite compelling, but the legends again were lacking. Why are there two Kd values in the upper panels of Fig. 4? It is not clear how the two Kds were extracted from the data based on the single line in the inset Scatchard plots. These two high and low affinity interactions are not even mentioned in the legend. How was the competition SPR executed to interpret the nature of the lower panels? This legend is exceptionally brief and it is difficult for the reader to interpret the data. Both Supplementary Fig. 4 and Supplementary Fig. 7 use 'mutant 3' and 'mutant 5', but only mutant 5 is cryptically described in the Fig S4 legend and it is hard to determine how they are acting as negative controls in the respective studies.

The FPOP data in Figs. 5, S5 and S6 are well presented and support the models for conformational changes of domains 1-5 and protection by ligand upon binding. The pH shift data also indicate major changes in accessible surface area, but is there a direct correlation with the modeled conformational changes? It is hard to tell based on the descriptions in the figures.

The potential role for a secondary binding site is also presented based on the FTMap, glycan modeling and FPOP analysis and seems a bit tenuous in the absence of additional data, but it is not a major concern for the manuscript. In Fig. 6A, what are the stick representations that are enclosed by the black dotted circle? They are not described in the legend.

Overall, the manuscript provides a compelling set of data that examines the structures and dynamics of the d1-5 segment of the CI-MPR. This is an important advance in our understanding of lysosomal enzyme targeting and CI-MPR function and is certainly worthy of publication in a high impact journal like Communications Biology. While the data in the manuscript are strong and appropriate, there are challenges with the data presentation and descriptions in the figures that are more confusing than necessary. As a result, there is high enthusiasm for the manuscript, but it will require revision to clarify numerous concerns in the legends, text, and, in some cases, the figures.

Reviewer #2 (Remarks to the Author):

Major points.

The manuscript "Allosteric regulation of lysosomal enzyme recognition by the cation-independent mannose 6-phosphate receptor" by Olson et al. reveals structures of the first five CRD domains of IGF2R and uses multiple complementary biophysical methods showing the mechanism of how this receptor is capable of tightly binding a long list of disparate client proteins. This manuscript includes the use of multiple orthogonal methodologies, including XRAY, TEM, SAXS, SPR, and HRPF, to show that the CRDs are dynamic with respect to each other under neutral pH conditions and cooperatively interact bivalently on client proteins. Overall, this manuscript is an important contributor to the field as it provides detailed insights into an essential and frequent disease-relevant cellular process. Other than the excellent and detailed materials and methods section, the rest of the manuscript may be unnecessarily long with many ancillary structural observations that although fascinating may not be essential for this forum; the succinctness of manuscript could be trebled without significant loss of impact.

Minor points.

Descriptors of relational discovery timelines should be excluded. The term 'high resolution' is relative; consider replacing with 'negative staining or transmission'. The method of hydroxyl radical protein footprinting (HRPF) is an important tool expertly used in this study to evaluate binding sites of client proteins as well as major conformational changes. As part of the manuscript submission process, I suggest that DNA sequences of all expression plasmids used in the study be deposited in the ncbi database. With respect to crystal growth, what were the drop/reservoir volumes, crystallization temperature, and crystal growth kinetics?

Reviewer #3 (Remarks to the Author):

Olson and Dahms et al. performed structural analysis of human CI-MPR N-terminal 5 domains by X-ray crystallography, SAXS, EM and hydroxyl radical footprinting. The effect of pH and ligand binding on protein conformation is one of the main interests in this study. Overall, their conclusion is supported by a lot of experimental and modelling data, that ligand binding to one site allosterically influences another ligand binding to a second. Nonetheless, several points need to be considered for improvement.

- Although many indirect evidences are shown to prove the allosteric regulation, direct evidence is lacking. pH-dependent conformational change was suggested by SEC and hydroxyl radical footprinting. Can they deny the possibility of pH-dependent non-specific binding to a SEC resin? Can pH affect the reactivity of hydroxyl radicals to the protein, especially to His? pH-dependent structural change (change in radius of gyration) can be analyzed by SAXS, which is essentially free from artifact. I think additional direct evidence is required to support their conclusion.

- I do not know possible reason why electron density of M6P was not observed in the crystal structures. Is there a possibility that lack of the electron density is simply due to the low affinity of M6P? Did N-glycan at N591 also bind to the adjacent molecule in the crystal structure at pH 7.0, as observed in the crystal structure obtained at pH5.5? Is there any structural difference between 1SYO and 6P8I in the ligand binding site? M6P is a small molecule and the binding seems to be characterized only by the binding site geometry in the CRD.

- Mismatching is found for text and figure legend. For example, Figure 1b shows one crystal structure but the legend says comparison of two crystal structures.

Response to Reviewers:

We thank the reviewers for their valuable comments and careful critique of the manuscript. We have addressed all concerns raised by the reviewers as detailed as follows.

Reviewer #1

This paper describes a complex series of structural, SPR binding, FPOP, and modeling studies on a multi-domain fragment of the cation independent mannose-6-phosphate receptor (CI-MPR) to examine domain-specific conformational changes that occur upon ligand binding and pH dependent ligand release. It is a complicated manuscript that extends prior data from the same lab on the structural basis of ligand interactions for the CI-MPR, a multidomain receptor that binds M6P-containing ligands using several of its MRH domains as well as interactions with other protein ligands with other domains. The authors have previously determined crystal structures of the bovine CI-MPR for domains 1-3 as well as domain 5 and others have determined structures of domains 11-14. Domains 3, 5, 9, and 15 have been shown to bind M6P-containing ligands. The present studies focus on crystal structures of domains 1-5 at pH 5.5 and 7.0. The authors then compare these structures with the prior domain 1-3 structures and perform SAXS, SPR, negative stain EM, and FPOP analysis at different pHs and in the presence of glycosylated ligands, and modeling of domain interactions.

While the manuscript is generally well-written, and the structural, FPOP, SPR, and modeling data are superb, there are several concerns regarding the data presentation that make the manuscript more confusing than necessary. The biggest challenge of the manuscript is the somewhat disconnected collection of data that is brought together to describe the nature of the domain interactions and conformational changes that occur during ligand binding and pH dependent binding and release of ligands. The manuscript starts with the presentation of X-ray structure data for domains 1-5 at 2 different pHs (pH 5.5 and 7) that show essentially the same protein structure at both pHs. This is a rather surprising result, since it is clear based on prior data from the CI-MPR and essentially all of the other data in the present manuscript that there are major conformational changes that occur during ligand binding and release for the domain 1-5 segments of the protein. The remainder of the manuscript is then focused on collecting data that indicate that the compressed domain assembly identified in the crystal structure does not reflect the dynamics of domain interactions of the protein in solution. The dichotomy between the identical, static X-ray structures at pH 5.5 and 7 versus significant structural differences in conformational dynamics in solution lead to a model for allosteric regulation of ligand interactions through global conformational changes in domain positioning.

- 1) *Probably the largest concerns in the manuscript are the relatively brief and non-descriptive figure legends that make interpretation of the figures and data a challenge.* We thank this reviewer for their comments and regret we did provide sufficient descriptions in the legends to allow the figures to be more easily followed. We have significantly enhanced the legends as detailed below as well as changing some of the figures to more clearly present our data to the reader.
 - a. *Panel 1B supposedly shows an overlay of the pH 5.5 and pH 7 structures, but it is almost impossible to even discern that there is even a grey cartoon representation for the pH 7 structure in the figure.* Coloring has been changed from grey to wheat to enhance contrast.
 - b. *Panel 1B has domain labels that are either hidden behind the protein structure or are far away from the respective domain. In most of the other panels the*

domains are labeled d1, d2, d3, etc, but in Panel 1B they are labeled 1, 2, 3, etc. Labels have been changed to d1-d5 to be consistent with other figures. Labels have also been repositioned for better viewing.

- c. *The dotted circle in Panel 1B is also described as a star in the legend.* This has been corrected.
- d. *The glycan associated with d5 in Panel 1B is also so light that it is difficult to see.* The figure has been re-rendered to darken the oligosaccharide.
- e. *In Panel 1C it is not clear from the figure legend why they are even shown.* This panel has been significantly redone combining panels c, d, and h. The significance of 1SYO has been elaborated upon in the figure legend.
- f. *It is also not clear until Panel 1H that the structures in Panel 1B and Panel 1C are essentially identical for d1 and d2. Are they supposed to be aligned based on those domains? If not, they really should be.* Information from panel 1h has been incorporated into panel 1C. Legend now states: Overlay of the C α atoms of domain 1 of PDB 1SYO
- g. *The M6P bound in Panel 1C appears to be a stick representation, but why are there stick representations of a glycan attached to d1? Is that a covalently attached glycan? It is not labeled in the figure nor described in the legend.* 1SYO structure has a glycan attached to domain 1 which has now been removed for clarity.
- h. *The differences in position and angle of d3 in Panel 1D are also hard to interpret since there is no frame of reference of an anchor position based on the positions of d1 or d2 in the panel to interpret the ~45° angle. The 45° label is also positioned partially behind the protein structure.* The frame of reference is now stated in the figure legend and accompanied by rotation angle in the figure. “ c, Overlay of the C α atoms of domain 1 of PDB 1SYO (M6P bound domain 3) (grey) with those of PDB 6P8I (unbound) (green). The upper right inset showing only domain 3 of each structure.”
- i. *Overall, Panels 1B-D are poorly described in the legend for a reader to take home the information content that the author intended.* Additional information about the figure has been added to the legend.
- j. *Similarly, the comparisons in Panels 1E and 1F are confusing since the authors do not use a similarly positioned domain d1 and d2 in the two panels. These two domains are the frame of reference for the change in position in d3. They should be oriented the same or the point of the figure is lost.* Panels E and F are now panel D. Although the reviewer requested that the orientation should be identical for these two images, a slight twist was necessary in order to clearly show the side-chains.
- k. *Fig. 1H is wonderful in its comparison of the position of domain d3 and should have been used for Panels 1B and 1C, except that the poor contrast in colors for domains d3 bound and d3 unbound (light green and not quite so light green) make it very difficult to tell that there are even two different d3s in the panel.* Panel H has been incorporated into panel C as suggested. Domain 3 of 1SYO has been changed from light green to grey for better contrast.
- l. *Panels 1I and 1J are an interesting approach for presentation for the domain interfaces, but why is the full surface of domain d3 shown in Panel 1I when only the interfaces are shown for the other domains in the panels?* These panels were eliminated when the figure was rearranged.

- m. *Overall, the entire figure legend needs to be a more expansive in description for what the authors want the reader to see. This expanded description is not present in either the legend or in the text of the Results section and the reader is left trying to interpret the panels in the figure without sufficient information.* We thank the reviewer for pointing out how to improve our manuscript. We have expanded the figure legends to better explain what is represented in each image.
- n. *For Fig. 2 and Supplementary Fig 3 it is clear that the initial crystallographic models do not fit the SAXS envelopes well (measures of fit should be included for all panels in Fig. 2 and S3) and the extension of domain d5 appears to improve the fit indicating that there is likely significant dynamics of the respective domains in solution. However, the take home message of Supplementary Fig. 3 was hampered by the lack of a legend for Panel E, where the extension of domain d5 allowed an improved fit into the SAXs envelope. This hypothesis was supported by the negative stain EM imaging (Fig. 3), but again more expansive figure legends with potential interpretations of the figure data would help the reader understand what they were looking for. The lack of detail and interpretation in the legends and/or the Results section makes it hard to digest the data in those figures.* The legend has been added for panel e. Measures of fit are listed next to model. Greater detail and explanations have been added to each legend (main text and supplementary) to address this concern.
- o. *The data in Fig. 4 and Supplementary Fig 4 using SPR and competitive SPR were quite compelling, but the legends again were lacking. Why are there two Kd values in the upper panels of Fig. 4? It is not clear how the two Kds were extracted from the data based on the single line in the inset Scatchard plots. These two high and low affinity interactions are not even mentioned in the legend. How was the competition SPR executed to interpret the nature of the lower panels? This legend is exceptionally brief and it is difficult for the reader to interpret the data.* The legend has been significantly expanded with it now explicitly stated how the K_D was calculated. "Inset graphs show Scatchard plots based on average RU value collected over 10 sec time intervals at the end of the association phase for each concentration of domains 1-5 (red bar). The calculated K_{Ds} (-1/slope) values for two binding events are listed and were determined from an average of four experiments." The legend for the competition experiments was expanded to say "Results from the accompanying competitive inhibition study is displayed as double reciprocal plots (**lower panels**) in which domains 1-5 protein (at 10, 15, 20, 30, 40, 50, 60, 70, 80, 90, 100, 110 or 120 nM concentration) was preincubated for 2 hrs with 0, 40, 80, or 100 nM PPT1 (as indicated in the lower-left panel) before being flowed over the three different lysosomal enzyme surfaces as indicated in the upper panel."
- p. *Both Supplementary Fig. 4 and Supplementary Fig. 7 use 'mutant 3' and 'mutant 5', but only mutant 5 is cryptically described in the Fig S4 legend and it is hard to determine how they are acting as negative controls in the respective studies.* The names of these mutants have been changed to reflect the amino acid mutation which is further explained in the methods section.
- q. *The FPOP data in Figs. 5, S5 and S6 are well presented and support the models for conformational changes of domains 1-5 and protection by ligand upon binding. The pH shift data also indicate major changes in accessible surface area, but is there a direct correlation with the modeled conformational*

changes? It is hard to tell based on the descriptions in the figures. We have now been able to include a comparison with the cryo-EM low pH structure of CI-MPR recently published by Wang et al. (ref. # 28). The comparison can be found in lines 1023-1037. A table of surface area comparisons can be found in Supplementary Fig 6b.

- r. *In Fig. 6A, what are the stick representations that are enclosed by the black dotted circle? They are not described in the legend.* The legend has been expanded to say “are shown (colored stick representation) on the model of domains 1-5 bound to a ligand in domain 3. Region of predicted small molecule interaction near secondary site proposed from analysis of FPOP data is circled in dashed black line.”

Reviewer #2

Major points.

The manuscript "Allosteric regulation of lysosomal enzyme recognition by the cation-independent mannose 6-phosphate receptor" by Olson et al. reveals structures of the first five CRD domains of IGF2R and uses multiple complementary biophysical methods showing the mechanism of how this receptor is capable of tightly binding a long list of disparate client proteins. This manuscript includes the use of multiple orthogonal methodologies, including XRAY, TEM, SAXS, SPR, and HRP, to show that the CRDs are dynamic with respect to each other under neutral pH conditions and cooperatively interact bivalently on client proteins. Overall, this manuscript is an important contributor to the field as it provides detailed insights into an essential and frequent disease-relevant cellular process.

- 1) *Other than the excellent and detailed materials and methods section, the rest of the manuscript may be unnecessarily long with many ancillary structural observations that although fascinating may not be essential for this forum; the succinctness of manuscript could be trebled without significant loss of impact.* We thank this reviewer for his/her careful critique of our manuscript and subsequent suggestions on how to improve upon the overall presentation of our work. Many details have been moved from the text to the corresponding figure legend. Although the level of succinctness has been improved in many sections, the overall length of the manuscript has not been dramatically reduced. This is due to the Editor's request for a discussion of Wang's cryo-EM structure that resulted in new figures and text in the revised manuscript.

Minor points.

- 2) *Descriptors of relational discovery timelines should be excluded. The term 'high resolution' is relative; consider replacing with 'negative staining or transmission'.* These descriptors have been removed along with 'high resolution'.
- 3) *I suggest that DNA sequences of all expression plasmids used in the study be deposited in the ncbi database.* The original DNA for the full-length CI-MPR was purchased from ATCC and the NCBI number is documented in the Methods section. Line 1343
- 4) *With respect to crystal growth, what were the drop/reservoir volumes, crystallization temperature, and crystal growth kinetics?* This information has been included in the methods section and reads “A crystallization hit was identified using Molecular dimensions JCSG-plus (100 mM HEPES, 30% Jeffamine ED-2003) after more than 1

week at 19° C (sitting drops with 1:1 ratio of protein (0.15µl) to well solution (60 µl in well).” lines 1367-1371, 1380-1381

Reviewer #3

Olson and Dahms et al. performed structural analysis of human CI-MPR N-terminal 5 domains by X-ray crystallography, SAXS, EM and hydroxyl radical footprinting. The effect of pH and ligand binding on protein conformation is one of the main interests in this study. Overall, their conclusion is supported by a lot of experimental and modelling data, that ligand binding to one site allosterically influences another ligand binding to a second. Nonetheless, several points need to be considered for improvement.

We thank this reviewer for his/her careful reading of our manuscript and insightful comments and questions.

- 1) *Although many indirect evidences are shown to prove the allosteric regulation, direct evidence is lacking.* The cryo EM structure of bovine CI-MPR provides a direct evidence for the allosteric behavior of CI-MPR. This is described in the text (1074-1083), and Figure 8 clearly demonstrates this point.
- 2) *pH-dependent conformational change was suggested by SEC and hydroxyl radical footprinting. Can they deny the possibility of pH-dependent non-specific binding to a SEC resin?* Interactions between proteins and resins are always a possibility. However, in this case FPOP results at pH 4.5 are in agreement with SEC data indicating interaction with the resin during SEC is not occurring. Additionally, our pH 4.5 data are in agreement with the cryo-EM structure recently published by Wang et al. This is discussed in the manuscript, lines 1023-1037.
- 3) *Can pH affect the reactivity of hydroxyl radicals to the protein, especially to His?* The pH effects on FPOP is discussed in lines 1011-1012.
- 4) *pH-dependent structural change (change in radius of gyration) can be analyzed by SAXS, which is essentially free from artifact. I think additional direct evidence is required to support their conclusion.* SAXS at pH 4.5 would be a viable option for additional data if the cryo-EM structure at pH 4.5 mentioned above had not been published.
- 5) *I do not know possible reason why electron density of M6P was not observed in the crystal structures. Is there a possibility that lack of the electron density is simply due to the low affinity of M6P? Did N-glycan at N591 also bind to the adjacent molecule in the crystal structure at pH 7.0, as observed in the crystal structure obtained at pH5.5?* Our explanation for this is presented in lines 423-428 “Although we were surprised not to find M6P in the binding pocket of domain 3 under conditions highly favorable for ligand binding (pH 7.0) by this receptor, the presence of the N-linked glycan near the binding site of domain 5 could perturb conformational equilibria between substructures stabilizing a domain arrangement most favorable for crystallographic packing.”

Lines 417-423: “Interestingly, both conditions show evidence of the N-glycan at N591 of a crystallographic neighbor partially occupying the carbohydrate- binding site of domain 5. The pH 5.5 structure had sufficient occupancy to allow carbohydrate refinement while the corresponding region in the pH 7.0 structure did not (Fig. 1b)”

- 6) *Is there any structural difference between 1SYO and 6P8I in the ligand binding site? M6P is a small molecule and the binding seems to be characterized only by the binding site geometry in the CRD.* Binding site differences are shown now in Fig. 1c, lower right panel.

- 7) *Mismatching is found for text and figure legend. For example, Figure 1b shows one crystal structure but the legend says comparison of two crystal structures.* Figure issues have been addressed and are outlined in responses to reviewer #1.

Summary of Figure Changes:

Fig. 1: The number of panels has been reduced from 9 to 4 to simplify the presentation. In panel a, all surfaces have been removed. Panel b has gained an insert showing electron density around the domain 5 carbohydrate binding site. The labeling of domains has been changed to be consistent with other figures. The large red circle around the *N*-glycan has been changed to black. Panel c has been changed to a composite figure with a lower right panel showing the differences in the binding pocket in the presence and absence of ligand as requested by a reviewer. Panel d, the coloring was changed in the right-hand panel and labels were added beneath the figure to aid the reader.

Table 1 was converted to the requested journal format and information removed was relocated to the methods section.

Fig. 2: Chi-squared values were added to panel a and the color of the cartoon was changed to red to match panel c.

Fig. 4: Linear curves were added to the upper inset graphs to improve clarity and red bars showing the origin of the data in the inset have also been added.

Fig. 5: Black arrow added for clarity to show possible domain relocation.

Fig. 7: Old panel d removed and replaced by new panel f representing our FPOP data mapped onto the recently published Wang cryo-EM model. The bar graph was reformatted to meet journal requirements. Panel d was added to compare our structure at pH 5.5 to the Wang cryo-EM structure determined at pH 4.5.

Fig. 8: New figure added to examine the recently published cryo-EM structure with IGF2 bound to domain 11 and the changes in the domain associations of domain 9 with IGF2 bound versus pH 4.5.

Supplementary Fig. 1: Yellow color was defined.

Supplementary Fig. 2: Four panels relocated to new Supplementary Fig. 7 and panel h has been removed. Table labels altered to provide more information to the reader.

Supplementary Fig. 3: A new panel e was added to show the comparison of our structure to the newly released cryo-EM structure at pH 4.5 of CI-MPR.

Supplementary Fig. 4: Figure panels were rearranged due to changes in the manuscript. Scatchard plots with associated linear fit lines were added along with the calculated K_D values.

Supplementary Fig. 6: A new panel b was added to compare the interfaces of 6P8I to 6UM1 (newly released cryo-EM structure at pH 4.5).

Supplementary Fig. 7: This figure was formally part of Supplementary Fig. 1 but has been relocated due to changes in the manuscript. Many labels have been added for clarity.

Supplementary Fig. 8: Graph titles for two lower left figures have been changed for clarity.

REVIEWERS' COMMENTS:

Reviewer #1 Says to the editor that the manuscript is acceptable in its current form.

Reviewer #2 (Remarks to the Author):

This manuscript provides biophysical evidence that the architecture of IGF2R, with 15 CRD domains, and specifically the first 5 CRDs, undergo significant re-organization and compaction upon ligand (PPT1) binding, results that are consistent with the the cryoEM structures published by Wang et al. at the time of this paper's submission. Moreover, the re-organization and compaction phenomenon not only provide the required valency of interaction with ligand but also induces allosteric changes that improves ligand binding (cooperativity). In the rebuttal, the authors addressed my question sufficiently. And although sufficiently addressing reviewer #1's concerns regarding the extend of legend details, they did not address reviewer #1's concerns regarding the order of results being presented, namely that the crystallography results, which did not support the hypothesis, should have been presented last not first. Importantly, the authors incorporated in this revised version a comparison and discussion of the cryoEM structures, significantly improving the manuscript.

Reviewer #3 (Remarks to the Author):

The manuscript is revised properly.